# The Radio-Frequency Transformer for Signal Separation

## Abstract

We study a problem of signal separation: estimating a signal of interest (SOI) contaminated by an unknown non-Gaussian background/interference. Given the training data consisting of examples of SOI and interference, we show how to build a fully data-driven signal separator. To that end we learn a good discrete tokenizer for SOI and then train an end-to-end transformer on a cross-entropy loss. Training with a cross-entropy shows substantial improvements over the conventional mean-squared error (MSE). Our tokenizer is a modification of Google's SoundStream, which incorporates additional transformer layers and switches from VQVAE to finite-scalar quantization (FSQ). Across real and synthetic mixtures from the MIT RF Challenge dataset, our method achieves competitive performance, including a 122x reduction in bit-error rate (BER) over prior state-of-the-art techniques for separating a QPSK signal from 5G interference. The learned representation adapts to the interference type without side information and shows zero-shot generalization to unseen mixtures at inference time, underscoring its potential beyond RF. Although we instantiate our approach on radio-frequency mixtures, we expect the same architecture to apply to gravitational-wave data (e.g., LIGO strain) and other scientific sensing problems that require data-driven modeling of background and noise.

## 1 Introduction

Many sensing and inference problems in the physical sciences can be cast as recovering a signal of interest (SOI) $\mathbf{s}$ from an additive mixture $\mathbf{y} = \mathbf{s} + \mathbf{b}$, where $\mathbf{b}$ is interference (or noise, or background, depending on context). There are many variations of this problem, and the one we are focused on here is the case where we have complete statistical description of the SOI, but only sample access to $\mathbf{b}$. Note that classical detection and estimation theory typically postulates a simple (often Gaussian) model on $\mathbf{b}$, but for many modern scenarios this modeling is too inaccurate.

Examples of this setting are abundant throughout engineering and sciences. Indeed, source separation is crucial in gravitational-wave detection, where strain data can be modeled as superposition of SOI (a chirp, in fact) immersed in nonstationary noise and learned representations complement matched filtering (Gabbard et al., 2018; George and Huerta, 2018; Ormiston et al., 2020), and collider physics at the LHC, where collision events are corrupted by pileup and per-particle or per-track tokenization underpins modern pileup mitigation and jet tagging (Bertolini et al., 2014; Komiske et al., 2017; Qu et al., 2022). We refer to Appendix A, Table 2 for a broader catalog of applications in natural science.

This paper focuses on the application in the radio-frequency (RF) domain, where SOI is a (scalar, or single-channel) digital communication signal and interference (which overlaps with SOI in frequency) may have rather diverse origins. Single-channel source-separation (SCSS) problem is pervasive in RF communications, radar and localization. The rapid growth of wireless devices under bandwidth constraints has congested the RF spectrum, making co-channel interference increasingly common. Consider Alice and Bob communicating over a shared channel while nearby Wi-Fi or 5G devices operate in the same band: extracting the SOI in this setting is a representative SCSS task.

Traditional estimators such as matched filtering and linear MMSE are performant only under restrictive distributional assumptions (e.g., Gaussian interference and jointly Gaussian sources), which are frequently violated in practice (Lapidoth, 2017). Supervised, data-driven approaches exploit the rich non-Gaussian structure of RF sources and interference and have demonstrated improved separation performance over classical pipelines (Lee et al., 2023; Lancho et al., 2025a); however, common convolutional designs rely on fixed-size inputs and very long receptive fields, complicating low-latency deployment when sequence lengths and timing vary. In contrast to currently used MSE-based training, we propose to first capture the underlying discretization of SOI via a learned tokenizer, and then train a source separation model with cross-entropy objective, built on a autoregressive transformer backbone (Vaswani et al., 2017). This approach makes our predictions more aligned with the final discrete metrics.

Through experiments on the MIT RF Challenge dataset (Lancho et al., 2025a), we demonstrate the competitive performance of our proposed model. A key metric for RF source separation is the bit error rate (BER), as it reflects communication reliability measured by recovery of the transmitted bits. We show that the proposed architecture, when trained to decode tokenized SOI representations, is able to achieve greater than $100\times$ reduction in BER in challenging 5G interference settings.

We further show that the RF transformer exhibits strong zero-shot generalization to additive white Gaussian noise (AWGN), a prevalent form of real world interference, achieving near optimal suppression despite having seen no such examples during training.

## 2 RF SOURCE SEPARATION BACKGROUND

Matched filtering (see Appendix B.1) remains a simple yet widely used method for interference mitigation, offering optimal performance under additive white Gaussian interference. However, its effectiveness declines in more complex interference scenarios, underscoring the value of modeling the rich structure of RF signals. Meanwhile, traditional approaches like maximum likelihood estimation (Shilong et al., 2007; 2008) depend on accurate statistical models, which are often unavailable or incomplete in practice, leading to degraded performance in real-world conditions (Lee et al., 2011; Chevalier et al., 2018).

When the statistical model is unknown, data-driven methods, especially those leveraging deep neural networks, have become popular for RF source separation, as they learn signal statistics directly from data. A commonly studied setting involves a single signal of interest (SOI) mixed with one interference source, modeled as

$$\mathbf{y} = \mathbf{s} + \kappa\mathbf{b}, \tag{1}$$

where $\mathbf{s}$ represents the SOI, and $\mathbf{b}$ is the interfering signal. In this setting, assuming unit power signals, we can quantify the relative levels of SOI power to interference power through the signal-to-interference ratio,

$$\mathrm{SIR}(\kappa) := \frac{1}{\kappa^2}. \tag{2}$$

One of the earliest works on end-to-end RF source separation (Lee et al., 2023) showed that directly applying supervised audio source separation methods discussed in Appendix B.2 yields suboptimal results due to the discrete nature of RF signals, long-range temporal dependencies, and overlap in both time and frequency domains. To address this, the authors introduced an enhanced Wave-U-Net architecture, which we will refer to as the UNet, with a wide initial convolutional kernel designed to capture signal-specific features like the cyclic prefix in OFDM. Subsequent work (Lancho et al., 2025a) extended the architecture to handle real-world signals, including over-the-air UAV and microwave emissions. They also proposed a WaveNet-inspired model using dilated convolutions to mimic wide kernels, which outperforms prior methods, particularly on challenging OFDM interference mixtures.

Separately, several novel architectures were introduced in the ICASSP 2024 SP Grand Challenge (Jayashankar et al., 2024) by benchmarking performance on the RF Challenge dataset consisting of various synthetic and over-the-air signal recordings. The approach in

(Henneke, 2024) improved reconstruction fidelity by adding a signal-matched autoencoder to the baseline WaveNet, fine-tuned to reduce mean squared error. The challenge winner (Tian et al., 2024), achieved state-of-the-art results on multiple mixtures by enhancing the WaveNet with learnable dilations and fine-tuning on synthetic data. Inspired by recent progress in audio source separation, Yapar et al. (2024) adapted the Demucs architecture (see Section B.2) to estimate the SOI waveform bits using maximum likelihood training, while Damara et al. (2024) integrated attention layers into a UNet to better capture long-range dependencies in the signal.

Recently, unsupervised approaches for RF source separation that leverage independent priors through diffusion models have gained significant attention. The method in (Zilberstein et al., 2023) introduces an algorithm for symbol detection in MIMO systems, which could potentially assist in signal recovery. In contrast, Jayashankar et al. (2023) present a novel optimization framework based on a modified posteriori (MAP) estimation via learned score function at different levels of Gaussian smoothing (obtained from a trained diffusion model), requiring no prior knowledge of the mixture signals.

## 3  PROPOSED ARCHITECTURE

Convolutional architectures are the dominant approach for RF source separation, leveraging the inductive biases of digital communication signals (see Section 2). While effective, these models rely on large receptive fields and struggle with variable-length mixtures and real-time processing.

Motivated by the success of transformers in language and vision tasks, we propose a transformer-based architecture for RF source separation that enables large-scale learning and autoregressive decoding. We start by providing an overview of our architecture and then explain each component in detail.

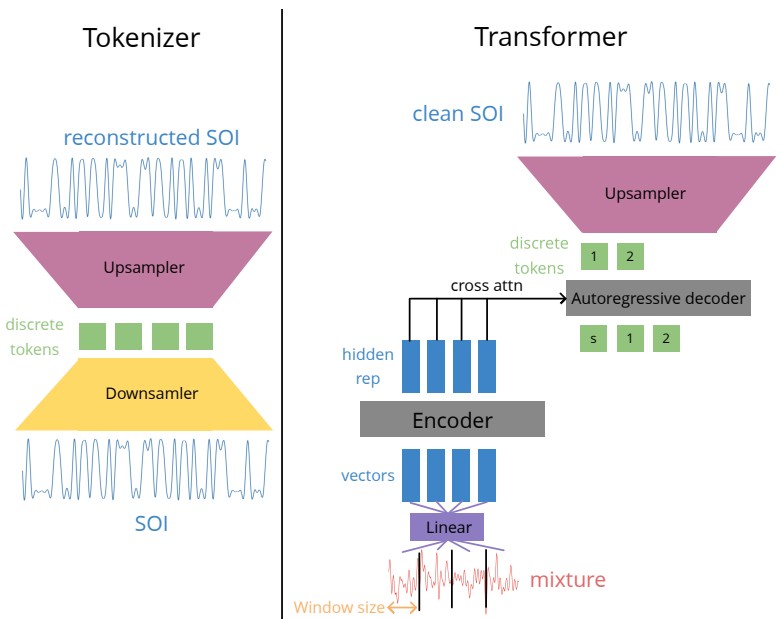

Figure 1: The schematic overview of the proposed architecture

### 3.1  ARCHITECTURE OVERVIEW

As shown in Figure 1, our architecture consists of two components: a tokenizer that learns discrete representations of the SOI, and a transformer that predicts a tokenized encoding of the SOI from a mixture. The tokenizer is implemented with an encoder-decoder architecture where the encoder maps the SOI $\mathbf{s} \in \mathbb{C}^N$ to a discrete-valued sequence $\mathbf{c} \in \{1, 2, \dots, k\}^L$ and

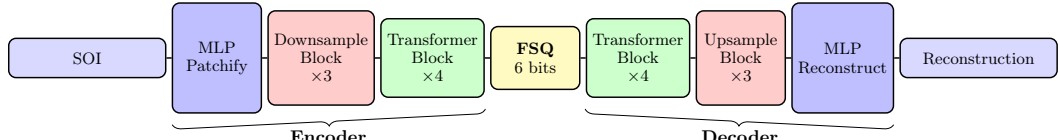

Figure 2: Overview of the SOI Tokenizer architecture. The main differences from the SoundStream architecture are: (i) additional Transformer blocks after downsampling and before upsampling; (ii) the use of FSQ instead of RVQ for discretization; and (iii) the omission of the discriminator network.

the decoder learns the reverse mapping back to the SOI. Here $k$ is the alphabet size defined by the total number of possible tokens. The encoded sequence length is $L = \lceil N/w \rceil$, where $w$ is the window size that controls the number of SOI samples that are compressed into one token. The tokenizer is trained by minimizing the MSE loss between the reconstructed and ground truth SOI waveforms.

For the transformer, we adopt an encoder-decoder architecture (Vaswani et al., 2017), where the mixture is processed by the encoder, and the decoder predicts the tokenized SOI waveform autoregressively. Following this, the pre-trained tokenizer's decoder converts the SOI tokens into a continuous waveform from which the underlying bits can be recovered using matched filtering.

Next, we describe these two components in more detail, starting with the tokenizer.

## 3.2 THE SOI TOKENIZER

Our tokenizer builds on the SoundStream encoder-decoder architecture originally developed for neural audio compression (Zeghidour et al., 2021), which uses a residual vector quantization (RVQ) module to produce discrete representations of input waveforms. However, directly applying this design to RF signals is suboptimal. To better capture the unique structure and statistical properties of RF data, we introduce several key modifications tailored specifically for RF signal tokenization.

Given the inherent discreteness of RF signals and to aid in training the transformer on practical sequence lengths, we are interested in further compression of the underlying information and hence consider an extremely low-bitrate setting for tokenization. To achieve this, we substituted RVQ with finite scalar quantization (FSQ) (Mentzer et al., 2023), which we found to work better for this low-bitrate setup. We will elaborate on this further in Section 4.4. Additionally, we found that for the QPSK SOI, adding extra transformer blocks before and after FSQ in the encoder and decoder respectively also leads to better validation loss. The full architecture of our tokenizer is illustrated in Figure 2, and we train it using an MSE reconstruction loss and we backpropagate through the FSQ module as in (Mentzer et al., 2023).

## 3.3 THE RF TRANSFORMER

With a trained tokenizer for the signal of interest (SOI) in place, we can proceed to implement our source separation model. The proposed architecture is an encoder-decoder transformer trained to predict the tokenized representation of the SOI $\mathbf{s}$ from a given input mixture waveform $\mathbf{y}$.

The first step in our pipeline embeds the mixture signal $\mathbf{y} \in \mathbb{C}^N$ into a sequence of continuous-valued vectors. The signal is divided into non-overlapping windows of length $w$, with additional context of $c_L$ samples to the left and $c_R$ to the right of each window. Each windowed segment is linearly projected into a $d$-dimensional embedding, resulting in an embedding matrix $\mathsf{Z} \in \mathbb{R}^{L \times d}$, where $L = \lceil N/w \rceil$ is the number of segments. Specifically, the $i$-th embedding $\mathbf{z}_i$ is computed from the segment spanning positions $w \cdot i - c_L$ to $w \cdot (i+1) + c_R$, with zero-padding applied when indices exceed the signal bounds. Real and

imaginary components of the complex-valued input are treated as separate input dimensions during projection.

The mixture embeddings are processed by a stack of encoder blocks, while the discrete tokens corresponding to the (partially) decoded SOI are fed through a stack of decoder blocks. Each block follows the standard Transformer architecture, comprising self-attention, normalization layers, a feedforward network, and residual connections. Additionally, each decoder block includes a cross-attention mechanism that conditions the SOI representation on the encoder's final output. Instead of standard sinusoidal positional embeddings, we adopt rotary positional embeddings (Su et al., 2024).

The RF transformer is trained via teacher forcing with cross-entropy loss. The training dataset is composed of mixture-SOI pairs, where the SOI is tokenized. When running inference on a new mixture, we decode the tokens of the SOI autoregressively and then use the SOI tokenizer's decoder to reconstruct the signal in the waveform domain.

## 4 Experiments

### 4.1 Experimental Setup

We evaluated our proposed architecture on four distinct mixture signals. Each mixture includes a QPSK SOI and is corrupted by a different real-world interference signal from the MIT RF Challenge dataset: EMISignal, CommSignal2, CommSignal3, and CommSignal5G. We provide more details regarding these datasets in Appendix F, Table 4. Our training setup closely followed the protocol outlined in the ICASSP 2024 SP Grand Challenge on RF source separation (Jayashankar et al., 2024).

Both the tokenizer and transformer were trained on waveform segments of length $N_{\text{train}}$. During training, we randomly sampled independent SOI and interference signals, cropping each to length $N_{\text{train}}$. This is representative of an *unsynchronized setting*, where the start of the SOI waveform may not align with the start of a QPSK symbol. As a result, direct decoding using MF without accounting for symbol offset will fail. Compared to the synchronized setup used in the ICASSP SP Grand Challenge, this setting is more challenging but also aids in augmenting the training data which is vital for transformer training.

To create the mixture we selected a random SIR from which we can compute $\kappa$ to define the mixture as in (1). In practice we also augmented the interference signal by multiplying it with a random phase offset. Due to limited dataset size of CommSignal2, we also applied additional transformations to the interference for this dataset, which we describe in the Appendix F.

When testing, we used the signal length $N_{\text{test}} = 40960$, which could be larger than $N_{\text{train}}$. To deal with this scenario, we selected a set of overlapping windows of size $N_{\text{train}}$ with stride $s$. We obtained an SOI estimate after decoding the tokens using the tokenizer's decoder and the final prediction for each sample in the prediced waveform is the average of all predictions from overlapping windows. Section 4.4 contains an ablation study on the choice of $s$. In our experiments, we typically choose $N_{train} = 2560$.

In addition to the experiments, where different models are trained separately on four available datasets, we consider the setup where one *Multi-type* model is trained to cancel all four interferences simultaneously. We describe the training procedure and evaluation results on this task in section 4.3.

### 4.2 Results

We tested the models on a separate test set with 50 SOI-interference pairs. We swept across 11 SIRs, ranging from $-30$ dB to $0$ dB with a step size 3 dB. For each SIR, we computed the average MSE of model predictions and the BER. We compared against the WaveNet and existing baselines from the ICASSP 2024 SP Grand Challenge (see Section 2).

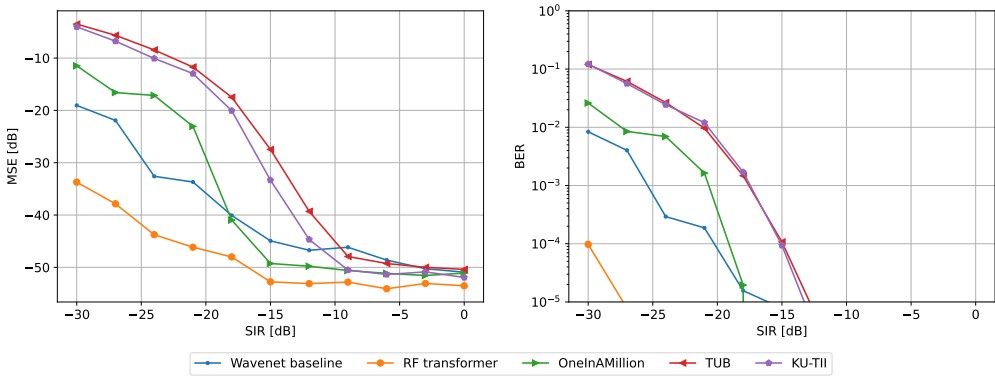

(a) Performance of various methods for separating QPSK and CommSignal5G interference.

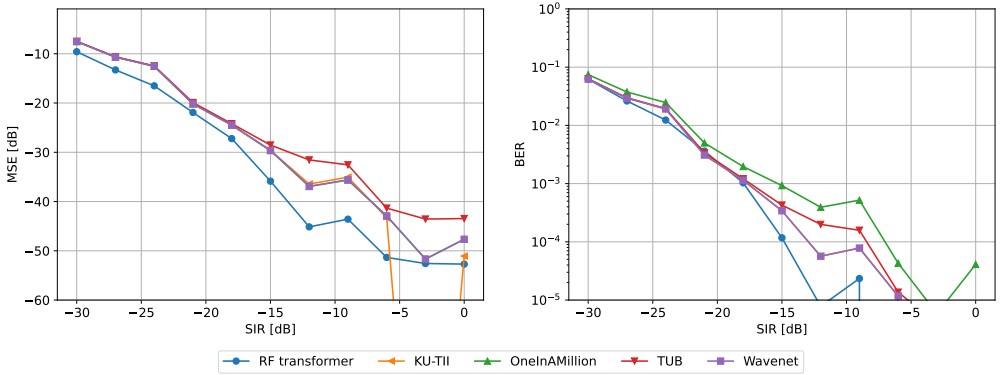

(b) Performance of various methods for separating QPSK and EMISignal interference.

Figure 3: Source separation performance for separating mixtures with CommSignal5G and EMISignal interference using different methods. In both cases our proposed architecture is highly competitive and surpasses most baselines across a wide range of SIRs.

Table 1: Performance of source separation methods on different interference types

| Method or team | MSE (dB) | | | | BER ($\log_{10}$) | | | |
|---|---|---|---|---|---|---|---|---|
| | CS2 | CS3 | CS5G | EMI | CS2 | CS3 | CS5G | EMI |
| RF transformer (**ours**) | -27.22 | -6.18 | **-46.32** | **-33.01** | -2.92 | -0.83 | **-4.91** | **-3.52** |
| RF transformer multi (**ours**) | **-28.71** | **-6.22** | -5.54 | -27.72 | **-3.07** | -0.92 | -0.86 | -3.05 |
| Wavenet | -24.14 | - | -39.43 | -28.92 | -3.05 | - | -4.23 | -3.33 |
| KU-TII | - | -6.04 | -30.17 | -29.07 | - | **-1.10** | -3.41 | -3.33 |
| OneInAMillion | -23.54 | -4.41 | -37.11 | -28.92 | -3.03 | -0.86 | -3.94 | -2.97 |
| TUB | -25.54 | -4.97 | -28.85 | -26.88 | -2.95 | -0.92 | -3.41 | -3.23 |

Table 1 summarizes the average performance of of our proposed method and baselines across the datasets. Note that for the KU-TII team, their outline performance on CommSignal 2 was excluded, due to leakage of the test set in the original challenge (Lancho et al., 2025b). For MSE, we take the average result in dB across SIRs, capping the MSE at -50 dB. We take the geometric mean of BER values, capping BER at $10^{-5}$.

Our method demonstrates strong performance across a range of interference types. Notably, as shown in Figure 3a it significantly outperforms baseline models on CommSignal5G and achieves state-of-the-art results for EMISignal as shown in Figure 3b. On mixtures involving CommSignals 2 and 3, our method attains state-of-the-art performance in MSE; for BER, it is state-of-the-art on CommSignal 2 and competitive on CommSignal 3. The total bit

errors across SIRs correlate with the average BER. In the case of 5G interference, our RF Transformer achieves an average BER of $9.59 \times 10^{-6}$, compared to $1.17 \times 10^{-3}$ for the Wavenet baseline — representing a $122 \times$ reduction in BER. Additional results are provided in the appendices; in particular, Appendix I contains preliminary results on real-time source separation.

### 4.3 MULTI-TYPE RF TRANSFORMER

Previously, we trained a different RF transformer for each different interference type. Here, we train a model to work in the setup where background can be composed of a mixture of multi-type interferences and Gaussian noise as well. To generate a training example, we sample SOI $s$ and four interferences $b_1, b_2, b_3, b_4$ from four available datasets. Let $\kappa$ be the coefficient that determines the SIR, and $(c_1, \ldots, c_5)$ be a uniformly sampled random point on a 5-dimensional sphere. In addition, we generate Gaussian noise $z$, where each component (real and imaginary) is sampled from $\mathcal{N}(0, 1)$. Then, our training mixture is

$$\mathbf{y} = \mathbf{s} + \kappa \left( c_5 \mathbf{z} + \sum_{i=1}^{4} c_i \mathbf{b}_i \right) \tag{3}$$

The goal of the model is still to recover $\mathbf{s}$ from this mixture. We find this training procedure to be more robust to overfitting issues compared to training with individual interference datasets with few samples, such as CommSignal2 and CommSignal3. Our model can deal with different kinds of interference simultaneously. The evaluation results of this model on original testsets are in table 1, row *RF transformer multi*. Similar to interference-specific models, we average the performance across SIR levels. Note that all the baselines were specifically trained for the respective datasets, while the Multi-type model is capable of operating on arbitrary interpolations of interferences.

We note that the Multi-type model outperforms the specialized RF transformer on CS2 and CS3, and results in weaker but still comparable performance on EMI signal. The only dataset that is hard for the Multi-type model is CommSignal5G, which benefits from specialized training. We also note that CommSignal5G is the only synthetic dataset among those four, which enables a specialized model to exploit potential invariants that are satisfied for this signal, while the model that always sees a noised interference might not be aware of them.

We also evaluate the Multi-type model on a test dataset generated according to (3). We use the matched filter as the baseline, as we do not expect specialized models to perform well on unknown structured interference. Results across SIR levels are shown in Figure 4.

### 4.4 ABLATION STUDIES

We validate the architectural design of the SOI tokenizer through a series of ablation studies. First, we compare FSQ and RVQ for tokenizing the SOI QPSK. We use FSQ with $b = 6$ bits and $[6, 4, 3]$ levels, and compare it with RVQ that employs two tokens, each encoded with 3 bits. All other hyperparameters are shared to ensure a fair comparison.

Second, we evaluate the effect of adding Transformer blocks within the tokenizer. We test both FSQ and RVQ with either 0 or 4 Transformer blocks. As shown in Figure 5a, FSQ consistently outperforms RVQ, and adding Transformer blocks further improves performance. All models are trained on waveforms of length 2560.

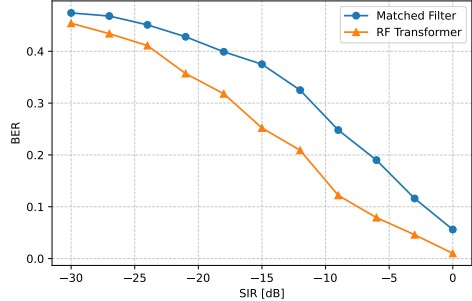

Figure 4: The BER comparison of matched filter and Multi-type RF transformer model on mixture dataset

Next, we investigate the impact on the performance of training on the signals of length 2560 instead of 40960. For both tokenizers we

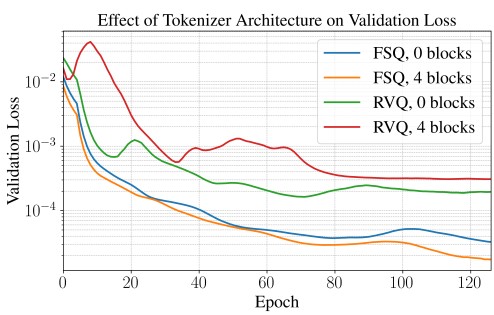 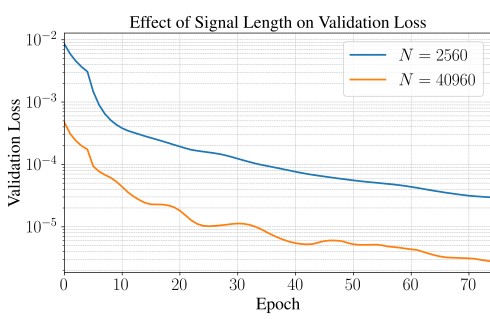

(a) Comparing quantizer and number of Transformer blocks.

(b) Comparing different input signal lengths for training.

Figure 5: Ablations studies evaluating key design choices for the SOI tokenizer. In (a), we show that combining FSQ with four transformer blocks yields the lowest validation loss among all configurations. In (b), we observe that tokenizer performance improves with longer input signal lengths.

use FSQ. We use 4 Transformer blocks for the signals of length 2560 and 1 block for the signals of length 40960. Although the performance drop is noticeable for length 2560 as seen in Figure 5b, we find that the Tokenizer remains adequate enough for the training of RF Transformer.

Finally, we also compare the performance of the CommSignal5G models with different window strides as shown in Appendix E, Figure 8. The results show that having more overlaps leads to better performance. However, this comes with the cost of increasing the number of windows on which we need to perform model inference.

### 4.5 Zero-Shot Performance for Mitigating Gaussian Interference

In this section, we study the generalization capabilities of the RF transformer on unseen mixtures at inference time, i.e., signal combinations not encountered during training. To this end, we pre-train an eight-layer transformer to separate mixtures of QPSK SOI and CommSignal2 (CS2) interference, using fully synchronized data. We then evaluate its performance on a foundational yet critical scenario: mitigating pure Gaussian interference.

It is worth noting that the most commonly used approach for interference mitigation, matched filtering (see Section B.1), is optimal under the assumption of additive white Gaussian noise (AWGN). As baselines, we compare the transformer's performance against both matched filtering and a linear minimum mean squared error (LMMSE) estimator, each of which has access to the true signal and interference statistics.

Our aim is to assess how the transformer model performs when varying levels of Gaussian noise are introduced by augmenting the mixture model as $\mathbf{y} = \mathbf{s} + \kappa_1 \mathbf{b} + \kappa_2 \mathbf{w}$. Here, $\mathbf{w} \sim \mathcal{N}(\mathbf{0}, \mathbf{I}_D)$ and $\kappa_2$ is a coefficient that controls the magnitude of the Gaussian noise. When $\kappa_2 = 0$, we recover the original mixture model from (1) used during training.

We can define an analogous quantity to the SIR, which is called the signal-to-interference-plus-noise ratio $\mathrm{SINR}(\kappa_1, \kappa_2) := 1/(\kappa_1^2 + \kappa_2^2)$, where we continue to assume that all the underlying signals have unit power. We also define the interference-to-noise ratio $\mathrm{INR}(\kappa_1, \kappa_2) := \kappa_1^2/\kappa_2^2$, which quantifies the relative strength of the structured interference $\mathbf{b}$ compared to the unstructured Gaussian noise $\mathbf{w}$.

Figure 6 summarizes the Transformer's denoising performance across a range of SINRs and INR values. Despite being trained exclusively on mixtures of QPSK and CommSignal2 waveforms, the model generalizes effectively to mixtures that include varying levels of Gaussian noise and degradation in performance is smooth as the INR decreases. It consistently outperforms the matched filtering baseline across all tested conditions and achieves lower BER than the LMMSE baseline in several regimes, particularly at high SINRs and moderate INR.

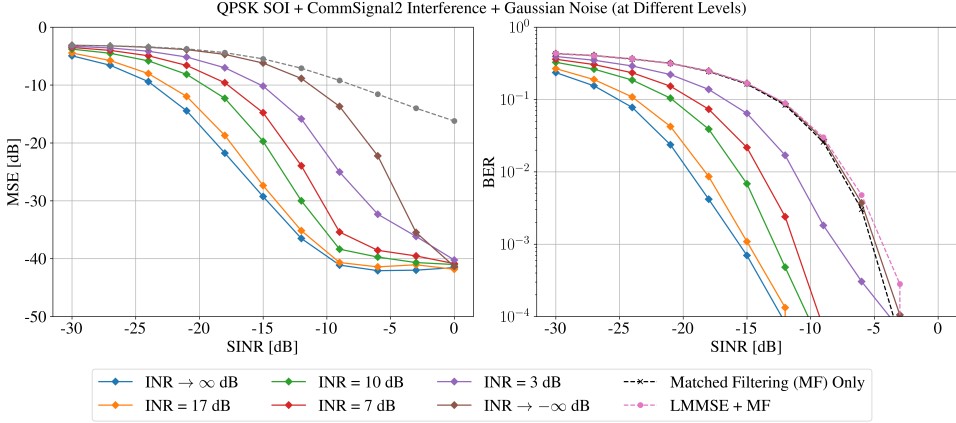

Figure 6: Denoising performance of a continuous transformer on synchronized QPSK signals corrupted by mixtures of CommSignal2 and Gaussian noise. Although trained without any Gaussian corruption, the model generalizes well and outperforms matched filtering and LMMSE at high SINRs.

Most notably, even when the interference is purely Gaussian (INR → −∞ dB), the transformer nearly matches the optimal BER achieved by matched filtering despite never being exposed to such noise during training. This behavior is notable, as Gaussian noise lacks the temporal and spectral structure of the training signals and lies entirely outside their distribution. The model's apparent robustness to such perturbations suggests that it is not merely memorizing waveform-specific patterns, but instead learning a more general and flexible representation of signal structure.

Beyond the CS2 study, we evaluate zero-shot Gaussian generalization across EMI, CommSignal3 (CS3), and CommSignal5G (CS5G). As INR decreases, performance degrades smoothly. In the pure-Gaussian limit (INR → −∞), models trained on EMI and CS3 match or closely approach the matched-filter baseline, whereas the CS5G model underperforms — likely reflecting differences in data origin (synthetic vs. recorded with ambient noise). A jointly trained Multi-type transformer also performs strongly when the structured interferer is recorded. Full results (MSE in dB and $\log_{10}$ BER versus INR) are provided in Appendix G, Table 5.

## 5 Concluding Remarks

In this work, we propose a novel Transformer architecture with autoregressive decoding for RF signal separation. To enable efficient training, we introduce a specialized tokenizer that discretizes RF signals, allowing the model to predict SOI tokens using a cross-entropy loss.

First, across diverse datasets, we demonstrate state-of-the-art performance in separating QPSK signals from CommSignal 2, 5G, and EMI interference types, and show competitive results against existing methods for CommSignal 3. Next, we train a *Multi-type* model that operates when multiple interference types are present simultaneously, achieving performance that is better than or comparable to interference-specific models (except for 5G). Finally, we show that our model exhibits zero-shot generalization to unseen mixtures at inference time.

**Reproducibility statement**. We provide experimental details in Appendix 4.1, including all model hyperparameters used for training, hardware configuration, and training regimen. We also include an anonymized codebase in the supplementary materials with a thorough README. The package contains dataset descriptions and preprocessing steps, scripts to train new models, and pretrained checkpoints for evaluation.

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

## A  Possible domains

In this section, we list some of the possible application domains for our method in Table 2.

| Domain | Mixture $y = s + b$) | Tokenization of $s$ | CE Target |
|---|---|---|---|
| RF communications | SOI + co-channel intf. | Constellation / codebook (FSQ) | Next-token |
| Gravitational waves (LIGO) | Chirp $h_\theta$ + noise | Quantized TF atoms / phase-increment | Template / token |
| LHC pileup mitigation | Leading-vertex + pileup | Per-particle/track (keep / PU / vertex) | Per-token labels |
| Seismology (phase picking) | P/S phases + ambient | {P,S,none} on time grid | Framewise |
| Neural spike sorting | Spikes + overlap/noise | (unit_id, binned $t$) tokens | Event sequence |
| Radio astronomy (FRBs) | Dispersed transient + RFI/sky | Dedispersed path / track tokens | Path / track |
| 21 cm cosmology | 21 cm + fg + instr. | Spectral / spatial codebook | Mask / fg |
| CMB component sep. | CMB + fg + noise | Patch / harmonic VQ tokens | Component label |

Table 2: Possible domains for our method. We list the field name; how its data fits our $y = s + b$ setup; the tokenization of the SOI $s$; and the target for cross-entropy training of an auto-regressive model.

## B  Source Separation background

In this section, we provide background on digital communications and a brief overview of deep learning methods for source separation.

### B.1  Digital Communication Signals

Digital communications deals with the transmission of bits by modulating a continuous waveform known as the carrier signal. At a high-level, before modulation, a digital communication signal can be represented in its complex baseband form as,

$$u(t) = \sum_{p=-\infty}^{\infty} \sum_{\ell=0}^{L-1} c_{p,\ell} \, g(t - pT_s, \ell) \exp\{j2\pi\ell t/L\}. \tag{4}$$

Groups of bits are mapped to symbols $c_p \in \mathbb{C}$ using a *digital constellation*, which assigns bit patterns to a finite set of complex values. These symbols are then combined into a continuous complex-valued waveform via (4), using a *pulse shaping filter* $g(\cdot)$ to limit bandwidth and reduce inter-symbol interference (Heath Jr, 2017, Sec 4.4.3). Although the waveform appears continuous, it still bears *underlying discrete structures* due to the finite constellation and deterministic filtering.

The constellation is largely defined by the number of bits grouped into a symbol. Common schemes include modulating two bits at a time (Quadrature Phase Shift Keying, or QPSK), or one bit at a time (Binary Phase Shift Keying, or BPSK). Additionally, multiple groups of bits can be transmitted in parallel by considering orthogonal sub-carrier waveforms, represented by by multiplication with multiple orthogonal complex sinusoids in (4). This is representative of Orthogonal Frequency Division Multiplexing (OFDM), inherent to many popular wireless standards such as 5G and WiFi.

To recover the bits at the receiver, one may adopt *matched filtering* (MF) (Lapidoth, 2017, Sec 5.8) before the estimation of the underlying symbols, and thereafter decode them back to bits. For commonly used pulse shaping functions, such as the root-raised cosine (RRC), the matched filter and pulse shaping filter coincide. We refer readers to (Lapidoth, 2017; Heath Jr, 2017; Goldsmith, 2005) for a more thorough exposition of the topic.

### B.2  Deep Learning for Source Separation

At a high-level, given a mixture signal,

$$\mathbf{y} = \kappa_1\mathbf{x}_1 + \kappa_2\mathbf{x}_2 + \cdots + \kappa_K\mathbf{x}_K, \, \mathbf{x}_i \in \mathcal{X}^N,$$

the goal of source separation is to recover the underlying components signals $\mathbf{x}_1, \mathbf{x}_2, \ldots, \mathbf{x}_K$. Above $\{\kappa_i\}_{i=1}^K$ are positive scaling coefficients that dictate the relative levels at which the signals interfere with each other.

Traditional source separation techniques often rely on simplifying assumptions and are limited in their expressive power. As a result, recent research has increasingly turned toward data-driven approaches powered by deep learning. For instance, end-to-end speech separation models that operate directly on time-domain waveforms using convolutional or recurrent architectures (Venkataramani et al., 2018; Stoller et al., 2018; Luo and Mesgarani, 2018; 2019; Tzinis et al., 2020) have demonstrated significant improvements over classical methods based on time-frequency masking (Wang and Chen, 2018) or non-negative matrix factorization (Yoshii et al., 2013).

Music source separation has also seen considerable advancements. While earlier methods primarily relied on spectrogram-based features (Liu and Yang, 2018) or recurrent networks (Takahashi et al., 2018), architectures like Demucs (Défossez et al., 2019) adopt a hybrid convolutional and recurrent model tailored for music signals. More recently, transformer-based models (Vaswani et al., 2017) which have set the state-of-the-art in autoregressive modeling, have been incorporated into source separation architectures, often via transformer blocks or cross-attention mechanisms, leading to further performance gains (Rouard et al., 2023; Lu et al., 2023). Recent architectures even adopt separate transformer encoders for encoding frequency and time information respectively as there is often only partial overlap in both domains. Even more recently, audio-visual source separation, wherein audio sources are separated by leveraging visual cues has grown increasingly popular (Afouras et al., 2018; Ephrat et al., 2018; Pian et al., 2024).

The source separation methods discussed above fall under the category of supervised approaches, which utilize paired datasets consisting of mixtures and their corresponding clean components. In contrast, unsupervised techniques have also been developed to address the same problem. For instance, BASIS separation (Jayaram and Thickstun, 2020) employs independent generative priors and performs image separation using annealed Langevin posterior sampling. Other methods take a different route by augmenting mixture data to generate synthetic training samples, thereby enabling unsupervised separation (Wisdom et al., 2020).

## C  Classical RF Interference Mitigation Techniques

In this section we briefly review the matched filtering and LMMSE estimation baselines that our used throughout this paper. The exposition below is inspired by discussions in (Heath Jr, 2017; Jayashankar et al., 2023; Lancho et al., 2025a).

### C.1  Matched Filtering

Matched filtering (MF) exploits knowledge about the signal to recover the transmitted symbols/bits. The basic principle involves filtering the received sampled RF waveform with a known filter called the "matched filter". The goal is to maximize the SINR at the filtered output, which consequently minimizes the error probability in the subsequent symbol detection when the noise is Gaussian.

Consider a QPSK signal represented with (4) corrupted with Gaussian noise which can be modeled as

$$y(t) = \sum_p c_p \, g_{\text{tx}}(t - pT_s) + w(t) \tag{5}$$

$$= g_{\text{tx}}(t) * \sum_p c_p \, \delta(t - pT_s) + w(t), \tag{6}$$

where $c_p$ are the symbols from a QPSK constellation , $*$ denotes the convolution operator, $\delta(\cdot)$ is the dirac delta function, and $w(t) \sim \mathcal{N}(0, \sigma_{\text{AWGN}}^2)$ is the additive noise in the observed signal, statistically independent of all $\{c_p\}$. Of particular interest in this formulation is the transmit pulse shaping function $g_{\text{tx}}(t)$, where we chose to use the RRC pulse shaping filter in this work.

At the receiver, we seek a receiver filter, $g_{\mathrm{rx}}(t)$, such that the filtered and sampled output

$$y_{\mathrm{filt}}(t) = \underbrace{g_{\mathrm{rx}}(t) * g_{\mathrm{tx}}(t)}_{:=g(t)} * \sum_p c_p \delta(t - pT_s) + g_{\mathrm{rx}}(t) * w(t) \tag{7}$$

$$y[n] = y_{\mathrm{filt}}(nT_s) = \sum_p c_p\, g((n-p)T_s) + \underbrace{\int w(\tau)\, g_{\mathrm{rx}}(nT_s - \tau)\mathrm{d}\tau}_{:=v[n]} \tag{8}$$

$$= \underbrace{c_n\, g(0)}_{:=y_s[n]} + \underbrace{\sum_{p \neq n} c_n\, g((n-p)T_s) + v[n]}_{:=y_v[n]} \tag{9}$$

would maximize the output SINR. In other words, we are looking to maximize

$$\mathrm{SINR} = \frac{\mathbb{E}\left[|y_s[n]|^2\right]}{\mathbb{E}\left[|y_v[n]|^2\right]} = \frac{\mathbb{E}\left[|c_n|^2\right]|g(0)|^2}{\mathbb{E}\left[|c_n|^2\right]\sum_{p \neq n}|g(pT_s)|^2 + \sigma_{\mathrm{AWGN}}^2 \int |G_{\mathrm{rx}}(f)|^2 \mathrm{d}f} \tag{10}$$

(where $G_{\mathrm{rx}}(f)$ is the Fourier transform of $g_{\mathrm{rx}}(t)$) via an appropriate choice of $g(t)$ — and thereby, $g_{\mathrm{rx}}(t)$. This can be done by finding an upper bound on the SINR that reaches equality for the appropriate filter choices. Ultimately, one such choice is $g_{\mathrm{rx}}(t) = g_{\mathrm{tx}}^*(-t)$ — termed as the *matched filter* — that leads to a maximized SINR. In the case of an RRC pulse shaping function (which is real and symmetric), the matched filter is also the same RRC function.

As part of the MF demodulation pipeline, the filtered output is sampled (as in (9)), and then mapped to the closest symbol. Finally, we can map these complex-valued symbols back to their corresponding bits to recover the underlying information. We use this as a standard demodulation/detection pipeline in our experiments.

Demodulation with matched filtering is optimal for waveforms in the presence of additive Gaussian noise. However, in our signal separation problem, we consider the presence of an additive interference, which is not necessarily Gaussian. Thus, exploiting the non-Gaussian characteristics of the interference would likely lead to enhanced decoding performance.

## C.2 LMMSE ESTIMATION

Recall that our observation model is

$$\mathbf{y} = \mathbf{s} + \kappa\mathbf{b},$$

where we assume $\mathbf{x}$ and $\mathbf{b}$ are zero-mean and that they are statistically independent. The linear minimum mean square error (LMMSE) estimator is the estimator $\widehat{\mathbf{s}} = \mathsf{W}_{\mathrm{LMMSE}}\mathbf{y}$, such that

$$\mathsf{W}_{\mathrm{LMMSE}} = \underset{\mathsf{W} \in \mathbb{C}^{T \times T}}{\arg\min}\, \mathbb{E}\left[\|\mathbf{s} - \mathsf{W}\mathbf{y}\|_2^2\right]. \tag{11}$$

In this case, the optimal linear transformation (in the sense of (11)) can be written as

$$\mathsf{W}_{\mathrm{LMMSE}} = \mathsf{C}_{sy}\, \mathsf{C}_{yy}^{-1} = \mathsf{C}_{ss}\left(\mathsf{C}_{ss} + \kappa^2 \mathsf{C}_{bb}\right)^{-1}$$

where $\mathsf{C}_{sy} := \mathbb{E}[\mathbf{s}\mathbf{y}^{\mathrm{H}}]$ corresponds to the cross-covariance between  and , $\mathsf{C}_{yy}$, $\mathsf{C}_{ss}$, $\mathsf{C}_{bb}$ are the auto-covariance of $\mathbf{y}$, $\mathbf{s}$ and $\mathbf{b}$ respectively. The second equality is obtained by statistical independence, thereby $\mathsf{C}_{sy} = \mathsf{C}_{ss}$, $\mathsf{C}_{yy} = \mathsf{C}_{ss} + \kappa^2 \mathsf{C}_{bb}$.

Since computing the covariance matrix can be expensive for long waveforms we implement a block-based LMMSE estimator by looking at short overlapping windows of the waveforms and computing the LMMSE estimate within these windows.

We remark that the LMMSE estimator is optimal if the components were Gaussian. However, as digital communication signals contain some underlying discreteness and undergo unknown time-shifts, these signals are typically non-Gaussian (and often, even far from Gaussian). Hence, better performance can generally be obtained through nonlinear methods.

## D  Experiment Configutation

In this section, we describe the network parameters and training hyperparameters to make our implementation more reproducible.

| Parameter | CommSignal2 | CommSignal3 | CommSignal5G | EMISignal |
|---|---|---|---|---|
| Train signal length | 40960 | 2560 | | |
| Encoder layers | 14 | | | |
| Decoder layers | 14 | | | |
| Embedding dimension | 768 | | | |
| Attention heads | 12 | | | |
| Window size | 16 | | | |
| Context size | $(16, 16)$ | | | |
| Token. channels | $[128, 256, 256]$ | $[256, 512, 512]$ | $[128, 256, 256]$ | |
| FSQ dimensions | $[6, 4, 3]$ | | | |
| Token. transformer blocks | 4 | | | |
| Patch channels | 8 | | | |
| Token. resnet count | 3 | | | |
| Optimizer | Adam (lr 0.0001, weight decay 0.01) | | | |
| Scheduler | ReduceLROnPlateau | | | |
| BF16 training | True | False | | |
| Batch size | 48 | 400 | 130 | 180 |
| GPU type | H100 | A100 | RTX A6000 | RTX 3090 |
| GPU count | 2 | 8 | 2 | 4 |
| Training time | 80 hours | 7 hours | 25 hours | 450 hours |

Table 3: Training setup for reproducing results.

## E  Additional Plots of Model Performance

The Figures 7 contains the performance plots of the model on other datasets from RF Challenge (CommSignal2 and CommSignal3).

In Figure 8 we compare the performance of the CommSignal5G models with different window strides.

## F  Dataset Description

Table 4 details the datasets used to train our models.

Figure 9 shows a representative sample from the SOI dataset, and Figure 10 shows samples from the four interference datasets used in our experiments. In all settings, we plot a segment of each sample in the time domain alongside a spectrogram illustrating its spectral content over time.

Table 4: Summary of the interference datasets used in our experiments.

| Interference | Dataset Type | Description | # Recordings | Recording Length |
|---|---|---|---|---|
| CommSignal2 | Recorded | Unknown | 100 | 43560 |
| CommSignal3 | Recorded | Unknown | 139 | 260000 |
| CommSignal5G | Synthetic | 5G OFDM signal | 149 | 230000 |
| EMISignal | - | Microwave Emission | 530 | 230000 |

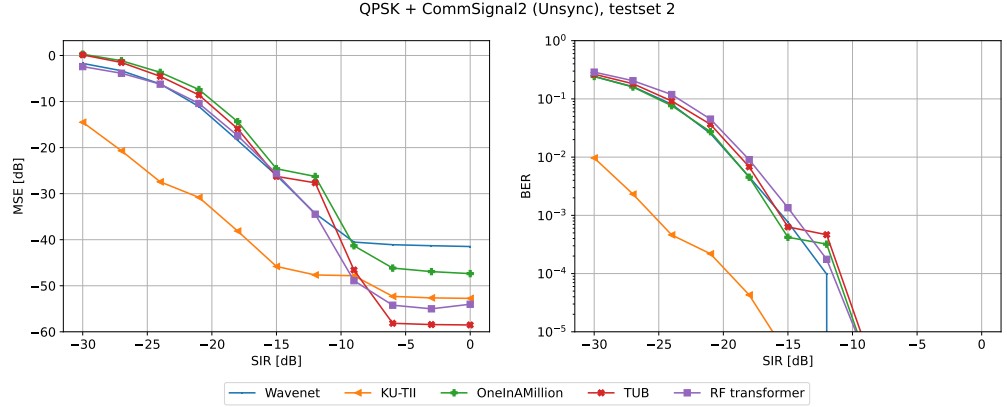

(a) Performance of various methods for separating QPSK and CommSignal2 interference.

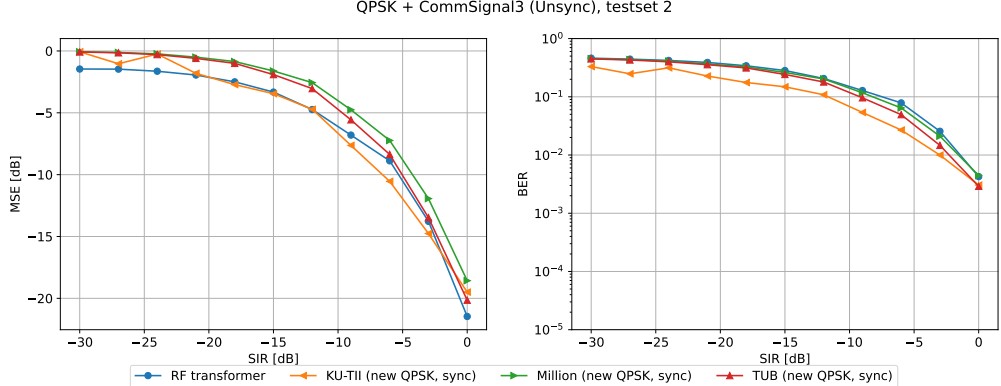

(b) Performance of various methods for separating QPSK and CommSignal3 interference.

Figure 7: Source separation performance for separating mixtures with CommSignal2 and CommSignal3 interference using different methods.

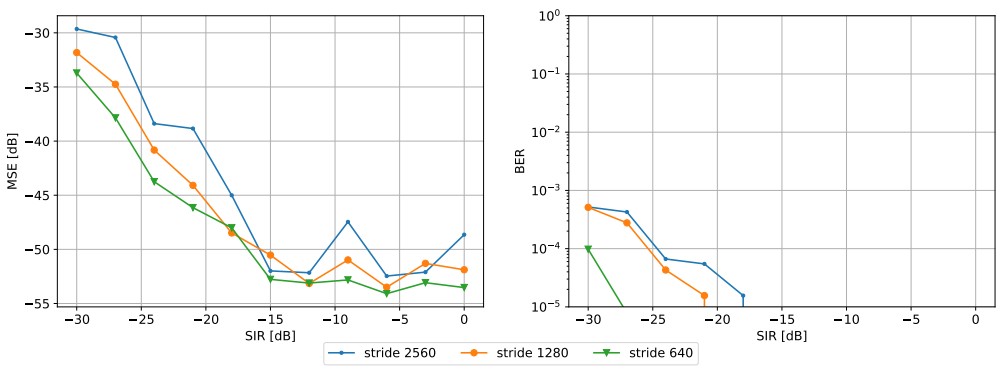

Figure 8: Effect of window stride on downstream source separation with 5G interference.

## F.1 Data Augmentation

For CommSignal2, because the interference dataset is small, we augment it with several data transforms. We list these transforms below.

1. **Random phase.** We generate a random complex number $\omega$ of absolute value 1, and multiply the interference by $\omega$.

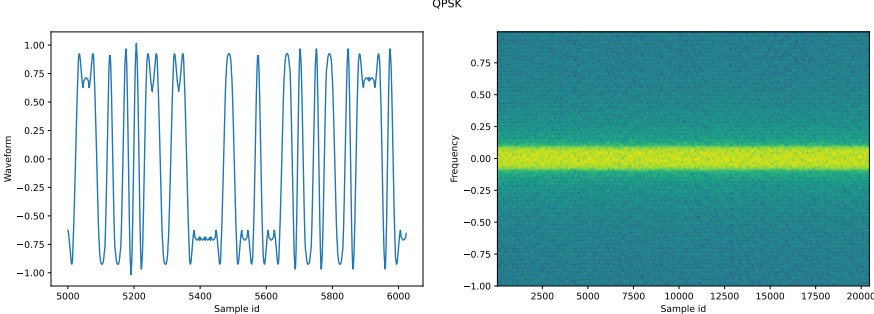

Figure 9: Real waveform and spectrogram of a sample from QPSK dataset

2. **Doppler shift.** We generate a random frequency $f$ with log uniform density between 200 and 2000. Then, the $k$-th sample is multiplied by $e^{\frac{ki}{f}}$.

3. **Shadow fading.** Unlike the two previous transforms, this transform modifies the magnitudes of the samples, and not their phases. This transform is parametrized by "sine magnitude" $a$, "noise magnitude" $b$, and frequency $f$. For each index $k$, we generate the amplification magnitude in dB, equal to $a\sin\left(2\pi\left(\frac{k}{f}+r\right)\right)\cdot a$, where $r$ is a uniform random number in $[0,1]$. These magnitudes are further augmented with Gaussian noise $\mathcal{N}(0,b^2)$. Finally, for a sample $z$ with amplification magnitude $p$, we augment the sample to $z\cdot 10^{\frac{p}{20}}$. For CommSignal2, we generate $f$ log-uniform between 200 and 2000, $a$ uniformly between 0 and 2, and $b$ uniformly between 0 and 0.01.

# G    Zero-Shot Performance for Mitigating Gaussian Interference

In this section, we provide additional results on the zero-shot performance of our transformer models for mitigating Gaussian interference.

## G.1    Gaussian-Noise Robustness Across Structured Interferences

In addition to the CommSignal2 (CS2) study in the main text, we evaluate zero-shot Gaussian generalization across three additional interference types: CommSignal3 (CS3), EMI, and CommSignal5G (CS5G). For each interference, models are evaluated over several INRs; we report both MSE (dB) and $\log_{10}$(BER) averaged across SINRs. We compare (i) a specialized transformer trained only on that interference, (ii) a Multi-type transformer trained jointly across all interferences, and (iii) a matched-filter baseline. Results are in Table 5.

Taken together, these results indicate that the RF transformer exhibits meaningful zero-shot generalization to noise, but that performance is sensitive to the interference's structure and origin; for synthetic datasets, explicitly varying noise during training may be necessary to obtain comparable robustness.

## G.2    Constellation Analysis

We begin by visualizing the predicted constellations at different SINR levels for both the LMMSE baseline and our transformer model (Figs. 11 – 12). At high SINR (e.g., 0 dB), both models produce tightly clustered points near ideal QPSK positions. However, at lower SINRs such as -15 dB, the LMMSE outputs become more dispersed, though they still generally fall within the correct quadrants. This suggests that symbol decisions remain largely accurate, which explains the model's low BER despite its poor MSE. In contrast, the transformer's constellation points remain sharply concentrated near normalized QPSK constellation points across all SINRs. This consistency reflects the model's end-to-end training on bit recovery

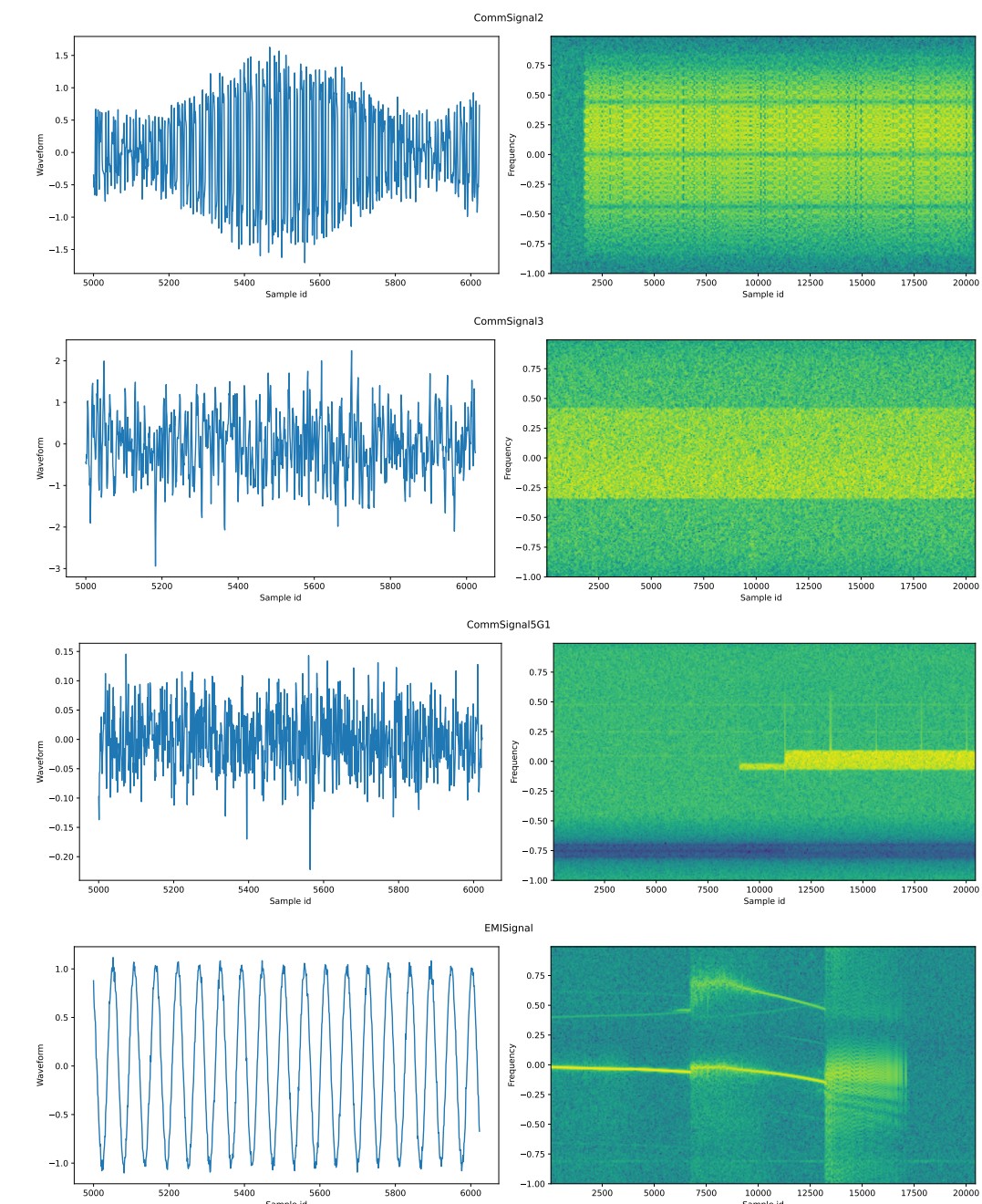

Figure 10: Real waveform and spectrogram of a sample from interference datasets

and suggests a strong internal representation of the modulation structure — even when evaluated on completely unseen interference.

## G.3 RAW WAVEFORM COMPARISON

To assess how each method reconstructs signal structure, we examine the real component of the raw predicted waveforms — prior to any demodulation or remodulation — for both LMMSE and the transformer (Figs. 13 – 14). At 0 dB, both methods track the true signal closely. However, at -3 dB, the LMMSE output begins to degrade: we observe amplitude damping, phase shifts, and increasing background noise. By -15 dB, the signal is almost

Table 5: Zero-shot Gaussian generalization across datasets. Columns are INR values; entries are averages over SINR.

| INR | MSE (dB) | | | | | log$_{10}$(BER) | | | | |
|---|---|---|---|---|---|---|---|---|---|---|
| | $\infty$ | 10 | 7 | 3 | $-\infty$ | $\infty$ | 10 | 7 | 3 | $-\infty$ |
| *EMI* | | | | | | | | | | |
| Single-type RF transformer | -33.01 | -23.17 | -20.58 | -16.54 | -11.66 | -3.52 | -2.54 | -2.35 | -1.93 | -1.43 |
| Multi-type RF transformer | -27.72 | -17.79 | -15.90 | -13.80 | -10.20 | -3.05 | -2.47 | -2.13 | -1.95 | -1.44 |
| Matched Filter | — | — | — | — | — | — | — | — | — | -1.43 |
| *CS2* | | | | | | | | | | |
| Single-type RF transformer | -27.22 | -23.58 | -21.54 | -17.86 | -12.00 | -2.92 | -2.58 | -2.37 | -2.04 | -1.48 |
| Multi-type RF transformer | -28.71 | -18.89 | -17.33 | -15.08 | -10.88 | -3.07 | -2.71 | -2.50 | -2.19 | -1.60 |
| Matched Filter | — | — | — | — | — | — | — | — | — | -1.60 |
| *CS3* | | | | | | | | | | |
| Single-type RF transformer | -6.18 | -6.32 | -6.62 | -7.01 | -11.22 | -0.83 | -0.85 | -0.88 | -0.92 | -1.55 |
| Multi-type RF transformer | -6.22 | -6.38 | -6.80 | -7.26 | -10.95 | -0.92 | -0.95 | -0.99 | -1.04 | -1.60 |
| Matched Filter | — | — | — | — | — | — | — | — | — | -1.62 |
| *CS5G* | | | | | | | | | | |
| Single-type RF transformer | -46.32 | -1.80 | -1.80 | -1.80 | -1.79 | -4.91 | -0.32 | -0.32 | -0.32 | -0.32 |
| Multi-type RF transformer | -5.54 | -4.52 | -4.32 | -4.34 | -10.62 | -0.86 | -0.76 | -0.74 | -0.74 | -1.45 |
| Matched Filter | — | — | — | — | — | — | — | — | — | -1.51 |

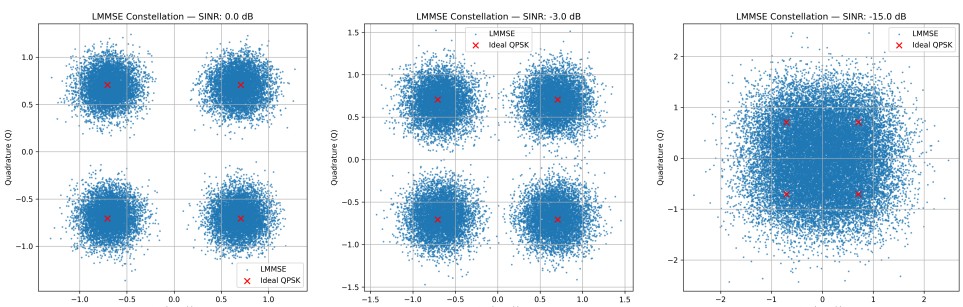

Figure 11: LMMSE constellation plots at 0 dB, -3 dB, and -15 dB SINR. Despite large variance at low SINR, the decoded symbols tend to remain within the correct QPSK quadrant.

unrecognizable, with erratic phase flips and flattened peaks. The transformer, by contrast, continues to capture the global waveform structure even at -15 dB, suggesting it can leverage prior structural knowledge to denoise in extreme settings. This resilience highlights its capacity for long-range contextual modeling, in contrast to the local and linear nature of LMMSE filtering.

### G.4 Waveform Recovery via Remodulation

Finally, we remodulate the predicted bitstreams to compare waveform fidelity after QPSK decoding (Figs. 15 – 16). For LMMSE, while bit-level decisions remain mostly correct, the remodulated waveform deviates in both amplitude and phase — especially under low SINR. This confirms the disconnect between LMMSE's low BER and high MSE: it finds the correct quadrant, but not the correct complex value. In contrast, the transformer's remodulated outputs are remarkably consistent with the true waveform, even at -15 dB, with phase and amplitude nearly intact. This suggests the transformer performs implicit denoising that aligns with modulation structure — recovering not just bits, but clean, waveform-consistent symbols.

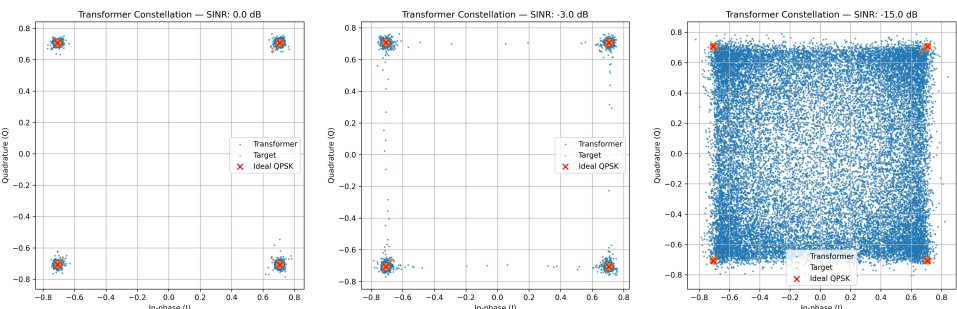

Figure 12: Transformer constellation plots at 0 dB, -3 dB, and -15 dB SINR. Outputs remain tightly clustered near ideal QPSK symbols even in highly noisy settings.

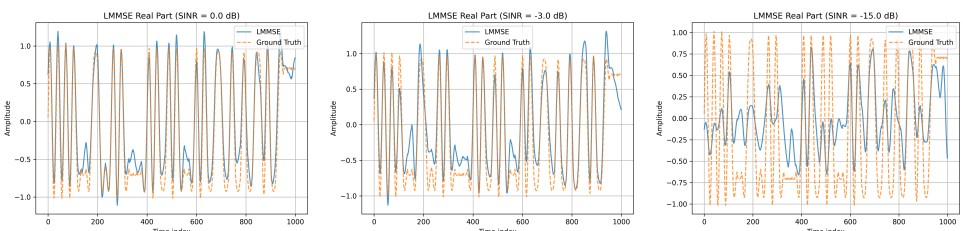

Figure 13: LMMSE real-valued waveform outputs at 0 dB, -3 dB, and -15 dB SINR, overlaid against ground truth. Note the increasing distortion and phase error at lower SINRs.

## H  RESOURCE EFFICIENCY METRICS

We report efficiency metrics for the RF Transformer and a WaveNet baseline on a 5G signal in Table 6. All measurements were performed on a 4xH100 node.

Although the RF Transformer consumes more resources than WaveNet, it consistently achieves superior performance because it can accommodate more parameters under comparable runtime constraints. Training the RF Transformer takes roughly three times longer, yet it yields error rates more than an order of magnitude lower than WaveNet. Moreover, the extensive literature on Transformer optimization offers promising avenues to further improve its efficiency and effectiveness. Finally, thanks to its ability to operate on a shorter signal window, the RF Transformer can achieve lower real-time source-separation latency.

| Attribute | RF Transformer | WaveNet |
|---|---|---|
| Parameters | 240M | 4M |
| Signal length | 2560 | 40960 |
| Batch size | 130 | 8 |
| Batch latency | 0.29s | 0.07s |
| Step count | 256,000 | 375,000 |
| Training time | 20.5 h | 6.5 h |
| Training throughput (samples/sec) | 1.1M | 4.6M |

Table 6: Comparison of RF Transformer and WaveNet training characteristics.

## I  TOWARDS REAL-TIME RF SIGNAL SEPARATION

In this section, we discuss real-time RF signal separation and the preliminary results we have achieved.

To perform inference in real time, we must make our models causal in time — that is, when separating the signal at any given moment, they should not look too far into the future. In

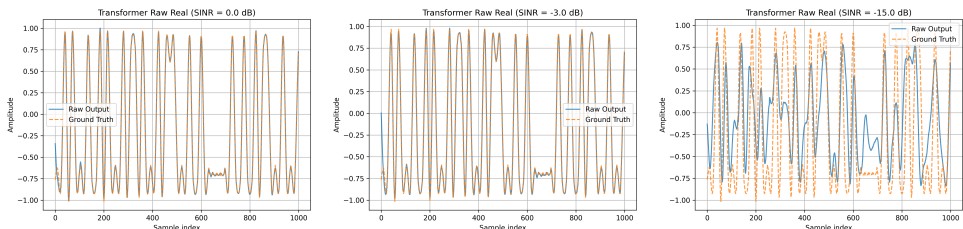

Figure 14: Raw transformer waveform outputs at 0 dB, -3 dB, and -15 dB SINR. Unlike the LMMSE results, the transformer is able to preserve signal structure even under high interference.

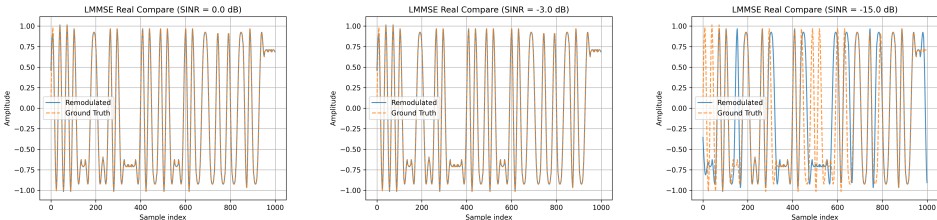

Figure 15: Real parts of remodulated LMMSE output vs. ground truth at 0 dB, -3 dB, and -15 dB. Despite correct bit decisions, amplitudes show mismatch at low SINRs.

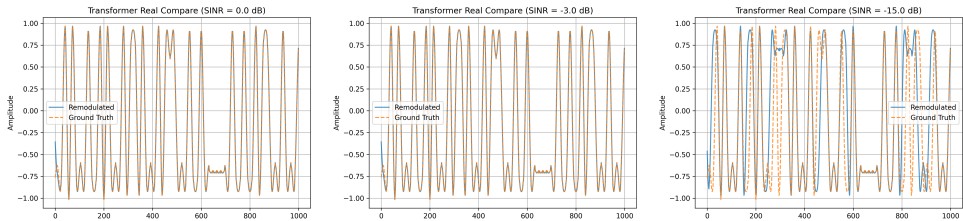

Figure 16: Real parts of remodulated transformer output vs. ground truth at 0 dB, -3 dB, and -15 dB. Transformer waveform closely matches ground truth at all SINRs.

most of our experiments, the RF Transformer and tokenizer are trained on signals of length $N = 2560$ and evaluated on signals of length $N = 40960$, which requires buffering only 2560 future samples at a time and thus already makes the architecture somewhat causal.

The RF Transformer's causality can be decomposed across four modules: the tokenizer, the encoder, the cross-attention block, and the decoder. However, our experiments show that making most of these components fully causal leads to training failure, so we allow a small lookahead in most cases.

To make the tokenizer causal, we must make each of its components causal: the convolutions in the upsampling and downsampling blocks, and the attention in the Transformer blocks. In Figure 17, we compare the performance of the causal tokenizer with its non-causal counterpart. We allow a one-token lookahead in each convolution and a three-token lookahead in the attention. To match the performance of the non-causal version, we increase the number of Transformer blocks from 4 to 8.

For the causal RF Transformer, we evaluate performance with a one-token lookahead in the cross-attention mechanism and zero lookahead in both the encoder and decoder — making the model as causal as possible while avoiding training failures. We also use the newly trained causal-in-time tokenizer. In Figure 18, we compare the causal RF Transformer with WaveNet and the non-causal RF Transformer on separating QPSK from CommSignal5G

interference. We observe a noticeable performance drop, which we aim to narrow in future work.

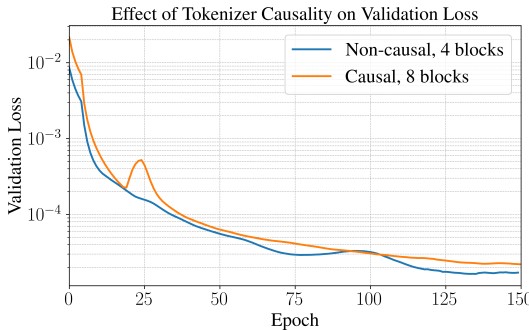

Figure 17: Comparison of tokenizer performance for the signal length of $N = 2560$. To achieve comparable performance in the causal setting, the number of Transformer blocks was increased from 4 to 8.

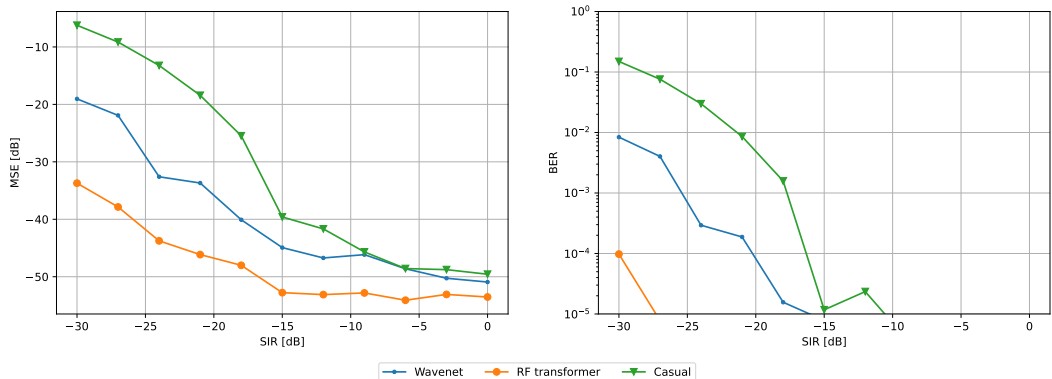

Figure 18: Performance of the original (non-causal) RF transformer, the proposed causal variant and (non-causal) WaveNet in separating QPSK and CommSignal5G interference. The proposed causal variant, though preliminary, shows competitive performance especially at high SIR.

