# OpenReview forum: "The Radio-Frequency Transformer for Signal Separation"
_ICLR.cc/2026/Conference — Submitted to ICLR 2026_

### Official Review · Reviewer_SA5U · 2025-10-24

**Soundness:** 3
**Presentation:** 1
**Contribution:** 2
**Rating:** 2
**Confidence:** 3

**Summary:**

The paper proposes a novel model architecture for signal of interest separation in congested radio frequency spectra. The approach consists of a VQ-VAE-based tokenizer that produces discrete representations of input waveforms and a transformer as source separation module estimating the discrete waveform representations. It uses finite scalar quantization instead of vector quantization for the VQ-VAE model and employs transformer blocks throughout both proposed network architectures. Extensive experiments are conducted, indicating a high performance and excellent generalization capabilities.

**Strengths:**

- The use of discrete representations for source signal representation appears to be original.
- The proposed method shows state-of-the-art performance in the signal of interest separation task and excellent generalization capabilities.

**Weaknesses:**

- I identified the use of a tokenizer to train a source separation model in the discrete domain as the main contribution of this paper. However, this design choice is not theoretically motivated nor evaluated. Furthermore, the authors claim that this model design improves training efficiency without providing any analysis.
- A large part of the paper focuses on the use of scalar quantization in the VQ-VAE and the employment of transformer blocks. However, these architectural improvements appear to be a minor contribution from a scientific POV.
- The ablation study only shows training loss curves which are hard to interpret. Instead, a comparison using the MSE and BER would should be presented.
- The performance gains for the test-sets QPSK+CommSignal2 and QPSK+EMI appear to be incremental

**Questions:**

Problems with the Presentation:
- Table 1 is not formatted correctly
- Inconsistent coloring in the result plots (makes it hard to follow)
- Figure 6 (left) what is the grey line representing?

---

> ### Author Response · Authors · 2025-11-21
>
> We thank the reviewer for taking time to read our paper and the provided feedback. In our response, we would like to address some of the issues raised in the review.
>
> > I identified the use of a tokenizer to train a source separation model in the discrete domain as the main contribution of this paper. However, this design choice is not theoretically motivated nor evaluated. Furthermore, the authors claim that this model design improves training efficiency without providing any analysis.
>
> In the RF setting the SOI is a QPSK, which is highly structured, and the final performance metric is BER. Framing the problem as predicting discrete SOI tokens and training with cross-entropy gives a surrogate that is much better aligned with BER than a waveform-level MSE objective. In particular, correct phase and symbol decisions are crucial, while small waveform distortions that preserve the bit sequence are largely irrelevant.
>
> Regarding “training efficiency,” our claim was that the optimization behavior is more stable and better targeted when the model predicts discrete symbols with CE loss.
>
> > A large part of the paper focuses on the use of scalar quantization in the VQ-VAE and the employment of transformer blocks. However, these architectural improvements appear to be a minor contribution from a scientific POV.
>
> We agree that using transformers and replacing RVQ with finite scalar quantization are architectural choices rather than fundamental theoretical advances, and we do not intend to oversell them as such.
>
> Our tokenizer is different from VAE. We chose FSQ over RVQ mainly because it simplifies codebook management and empirically worked better in our experiments on these RF datasets. Likewise, we chose transformer blocks as a backbone because they handle long-range dependencies and conditioning on the interference more naturally than purely convolutional designs.
>
>
> > The ablation study only shows training loss curves which are hard to interpret. Instead, a comparison using the MSE and BER would should be presented.
>
> We agree that training loss curves are not the most interpretable way to present ablations, especially for a task where BER is the main evaluation metric. However, due to compute and time constraints, it was not feasible to train everything end-to-end(tokenizer and transformer), so we limited ourselves to only retraining the tokenizer.
>
> > The performance gains for the test-sets QPSK+CommSignal2 and QPSK+EMI appear to be incremental.
>
> For CommSignal2, we achieved the top performance if we exclude KU-TII team result, which exploited the test data leakage.
> For the EMI, we achieved the best performance as well.
>
> > Table 1 is not formatted correctly
>
> We thank the reviewer for pointing this out. We will fix the formatting of Table 1.
>
> > Inconsistent coloring in the result plots (makes it hard to follow)
>
> We agree and will fix the color scheme so that each model has a consistent color across all panels.
>
> > Figure 6 (left) what is the grey line representing?
>
> The grey line represents the matched filtering baseline

---

> > ### Comment · Reviewer_SA5U · 2025-11-26
> >
> > I thank the authors for their detailed answer. I have read the other reviews and the responses to them as well.
> > I keep my score.

---

### Official Review · Reviewer_Cwjh · 2025-10-31

**Soundness:** 3
**Presentation:** 3
**Contribution:** 3
**Rating:** 6
**Confidence:** 3

**Summary:**

In this paper the authors propose the radio-frequency transformer, a method for performing source separation of (complex) scalar signals of interests (SOI) linearly perturbed by an interference, specifically targeting QPSK (a digital encoding scheme) signals in wireless communications. To this end they propose a conditional generative framework employing (i) a finite-scalar quantization [1] autoencoder based on Soundstream [2] which acts as a tokeniser on the continuous signal (ii) an autoregressive transformer conditioned on the mixed (perturbed) signal, modelling the tokens of the SOI. At inference time they auto-regressively decode the SOI given the conditional mixture, and then the discrete bit sequence can be obtained via matched filtering. The authors showcase state-of-the-art results on common datasets in the domain (CommSignal2,  CommSignal3, CommSignal5G, EMISignal) training a different model for each one, and additionally experiment with a multi-type single model on the four type of interferences combined (showing good results except on CommSignal5G).

### References
- [1] Mentzer, F., Minnen, D., Agustsson, E., & Tschannen, M. Finite Scalar Quantization: VQ-VAE Made Simple. In The Twelfth International Conference on Learning Representations.
- [2] Neil Zeghidour, Alejandro Luebs, Ahmed Omran, Jan Skoglund, and Marco Tagliasacchi. Soundstream: An end-to-end neural audio codec. IEEE/ACM Transactions on Audio, Speech, and Language Processing, 30:495–507, 2021.

**Strengths:**

The method showcases very good empirical results on common benchmarks in the wireless communications domain, surpassing current state-of-the-art models of the latest ICASSP challenge on wireless signal source separation. I also appreciate the effort put into making the model real-time, since this is the typical real-world use-case. Finally, the experiments on additive Gaussian intereference and the ablations are well received, showcasing improvement over classic baselines such as matched filtering and LMMSE.

The organisation and writing is smooth and the paper is not difficult to follow (maybe except for the details on QPSK bit encoding; I appreciate the appendix detailing those details for someone familiar with signal processing and deep learning but not familiar with the sub-field).

**Weaknesses:**

My thought reading this paper is that it would rank low on novelty since variants of the proposed method (conditional generation via autoregressive transformers for signal processing) is becoming more ubiquitous, for example in audio source separation [1] and accompaniment generation [2]. Also [3] should be mentioned as being the first method to perform source separation in a quantised autoencoder domain via autoregressive transformers. Nevertheless in the application domain of wireless comunications the method is novel as the use of finite-scalar quantization in this type of task.

I think the best thing for the work would be to be published in an IEEE journal given its engineering application focus, since ICLR is more oriented to novel methodologies in deep learning. Still, I am not against either for acceptance, given the overall good empirical results and writing, thus my recommendation score is 6.

### References

- [1] Yin, X., Peng, X., Jiang, X., Xiong, Z., & Lu, Y. (2025). Text-Queried Audio Source Separation via Hierarchical Modeling. arXiv preprint arXiv:2505.21025.
- [2] Rouard, S., San Roman, R., Adi, Y., & Roebel, A. (2025, April). MusicGen-Stem: Multi-stem music generation and edition through autoregressive modeling. In ICASSP 2025-2025 IEEE International Conference on Acoustics, Speech and Signal Processing (ICASSP) (pp. 1-5). IEEE.
- [3] Postolache, E., Mariani, G., Mancusi, M., Santilli, A., Cosmo, L., & Rodolà, E. (2023, June). Latent autoregressive source separation. In Proceedings of the AAAI Conference on Artificial Intelligence (Vol. 37, No. 8, pp. 9444-9452).

**Questions:**

- Line 041 to line 045: This phrase is too long containing many logical connectors (ands and ors that) over which is difficult to put
precedence.
- Line 054: It is not clear what is the problem with classic convolutional designs: I don't understand where they rely on fixed-sized inputs, during training or inference time? This should be the case for Transformers as well. Also convolutions are shift invariant so one could generalize to different sizes at inference time.  Is the problem with long receptive fields the fact that it would increases the latency at inference time? Also jumping to the real-time section of the paper, one does not store key-value cache over past windows to make it not scale quadratically in memory?
- Line 101: the cyclic prefix in OFDM is not explained
- Line 212:  What is the additional context $c_L$ ($c_R$)?
- Line 236: Maybe use we trained instead of evaluate, because evaluation is a synonym of testing.
- Line 237: Please send the reader to the appendix for the definition of QPSK, since it is not common knowledge except of the wireless communications field
- Line 244: I cannot understand what is the unsynchronized setting? Shouldn’t the SOI be encoded via QPSK? Why they are two different things?
- Line 256: "Predicted waveform"
- Figure 3: The authors should use the same colors for same models
- Line 327: In the real-time section one should report RTF for the models involved
- Line 358: "we do not expect specialized models to perform well on unknown structured interference.": This should be tested empirically

---

> ### Author Response · Authors · 2025-11-21
>
> We thank the reviewer for taking the time to read our paper and for the provided feedback. In our response, we would like to address some of the issues raised in the review.
>
> > My thought reading this paper is that it would rank low on novelty since variants of the proposed method (conditional generation via autoregressive transformers for signal processing) is becoming more ubiquitous, for example in audio source separation [1] and accompaniment generation [2]. Also [3] should be mentioned as being the first method to perform source separation in a quantised autoencoder domain via autoregressive transformers. Nevertheless in the application domain of wireless comunications the method is novel as the use of finite-scalar quantization in this type of task.
>
> We appreciate the reviewer pointing us to [1–3]. We were not aware of all of these references during our initial literature review and agree that they are relevant. All these works share the high-level idea of operating in a learned latent space and using autoregressive transformers for conditional generation. However, our focus and setting are different in several ways.
>
> We operate in the RF domain with QPSK signals and evaluate using BER, which is a system-level metric rather than a perceptual one. Our tokenizer is based on Finite Scalar Quantization, is fully pre-trained and then frozen. It serves as a post-processing step to produce discrete targets for the transformer, rather than being jointly adapted as part of generation. The transformer is conditioned directly on the RF mixture and trained to predict SOI tokens in a way that is tightly coupled to the matched-filter-based demodulation pipeline.
>
>
> > I think the best thing for the work would be to be published in an IEEE journal given its engineering application focus, since ICLR is more oriented to novel methodologies in deep learning. Still, I am not against either for acceptance, given the overall good empirical results and writing, thus my recommendation score is 6.
>
> We understand and appreciate this perspective. Our motivation for submitting to ICLR is that, beyond the application, the work demonstrates that recasting RF source separation as a discrete sequence prediction problem with a tokenizer + transformer pipeline can yield strong gains over standard regression-based architectures, and leads to robust zero-shot generalization in a challenging non-audio domain. We agree that the work is also a strong fit for an IEEE venue, and we view our contributions as sitting at the interface between deep learning methodology and wireless communications.

---

> > ### Author Response · Authors · 2025-11-21
> >
> > > Line 041 to line 045: This phrase is too long containing many logical connectors (ands and ors that) over which is difficult to put precedence.
> >
> > We agree with this comment, we will split this sentence into two shorter sentences to improve readability.
> >
> > > Line 054: It is not clear what is the problem with classic convolutional designs: I don't understand where they rely on fixed-sized inputs, during training or inference time? This should be the case for Transformers as well. Also convolutions are shift invariant so one could generalize to different sizes at inference time. Is the problem with long receptive fields the fact that it would increases the latency at inference time? Also jumping to the real-time section of the paper, one does not store key-value cache over past windows to make it not scale quadratically in memory?
> >
> > Thank you for pointing out that our original wording was unclear. Our main concern with purely convolutional designs is the trade-off between receptive field length and latency / efficiency when modeling long RF sequences in a causal, streaming setting. Achieving a sufficiently long receptive field with convolutions alone either requires very deep networks or large effective kernel sizes/dilations, which can increase latency and memory usage at inference.
> >
> >
> > > Line 101: the cyclic prefix in OFDM is not explained
> >
> > We agree, and will add a short explanation describing what the cyclic prefix in OFDM is.
> >
> > > Line 212: What is the additional context ()?
> >
> > “Additional context” refers to extra samples of the mixture signal included before and after the central window from which SOI tokens are predicted. This provides the model with boundary information around the prediction segment. We will clarify this explicitly in the text.
> >
> > > Line 236: Maybe use we trained instead of evaluate, because evaluation is a synonym of testing.
> >
> > We agree and will change “we evaluate” to “we trained” at this line.
> >
> > > Line 237: Please send the reader to the appendix for the definition of QPSK, since it is not common knowledge except of the wireless communications field
> >
> > We agree. We will add an explicit reference to the appendix where QPSK and the bit mapping conventions are described.
> >
> > > Line 244: I cannot understand what is the unsynchronized setting? Shouldn’t the SOI be encoded via QPSK? Why they are two different things?
> >
> > In the “unsynchronized” setting, the SOI is still encoded via QPSK, but the symbol alignment between the SOI and the receiver is unknown: there is an unknown time shift (up to one symbol) between the transmitter’s symbol boundaries and the receiver’s discretization. This leads to a mismatch between the nominal symbol grid and the actual positions of the QPSK symbols. We will clarify this in the main text.
> >
> > > Line 256: "Predicted waveform"
> >
> > We will change the wording to "predicted SOI".
> >
> > > Figure 3: The authors should use the same colors for same models
> >
> > We agree and will fix the color scheme so that each model has a consistent color across all panels.
> >
> > > Line 327: In the real-time section one should report RTF for the models involved
> >
> > We thank the reviewer for this suggestion. We will report RTF for our main models on a representative hardware configuration. This will complement our discussion of causality and sequence length, and make the real-time claims more concrete.
> >
> > > Line 358: "we do not expect specialized models to perform well on unknown structured interference.": This should be tested empirically
> >
> > We agree that this claim should ideally be supported by empirical evidence. Our current results partly touch on this point by comparing a single model trained on multiple interference types to specialized models trained on individual interferences. However, we did not systematically evaluate specialized models on out-of-distribution structured interferences.

---

> > > ### Comment · Reviewer_Cwjh · 2025-11-26
> > >
> > > I thank the authors for their detailed answer. I keep my score.

---

### Official Review · Reviewer_FwN4 · 2025-10-31

**Soundness:** 3
**Presentation:** 3
**Contribution:** 2
**Rating:** 4
**Confidence:** 4

**Summary:**

This paper introduces a Transformer-based architecture that recasts the problem of RF signal separation from a continuous regression task to a discrete sequence-to-sequence problem. The key idea is to first train a specialized tokenizer, leveraging finite scalar quantization, to learn a compact, discrete representation of the clean signal of interest (SOI). Then, an encoder-decoder Transformer is trained to predict this tokenized sequence from a noisy, mixed input. This token-based approach allows the model to be trained directly with a cross-entropy objective, which is far better aligned with the ultimate goal of minimizing bit-error rate (BER) than the conventional MSE loss used in prior work. The results are pretty compelling, showing a massive reduction in BER of over 100-fold in the challenging 5G interference case, as well as demonstrating zero-shot generalization to Gaussian noise, suggesting the model learns a robust underlying representation of the signal structure rather than just overfitting to specific interference types.

**Strengths:**

1. The paper steps outside the well-trodden domain of audio source separation and applies modern sequence-to-sequence modeling to the more constrained problem of RF signal separation. The domain is very intriguing and the success is measured by the unforgiving metric of Bit Error Rate (BER), not perceptual audio quality. By successfully adapting these advanced architectures to this field, the authors bridge a critical gap between mainstream deep learning and a specialized engineering domain with direct, tangible implications for wireless communications.
2. Perhaps the most important empirical result out of this paper is the model's really good zero-shot generalization to Additive White Gaussian Noise (AWGN). The Transformer was only ever trained on highly structured, real-world interference, yet when the input is disturbed with the unstructured Gaussian noise, it performed nearly as well as the theoretically optimal matched filter. This is a powerful demonstration that the model didn't just learn to "invert" a few specific interference types but it learned a deep and robust representation of the underlying structure of the QPSK signal itself.

**Weaknesses:**

The paper has some weaknesses and I will try to write them down with decreasing order of significance.

1. The authors should consider placing their contributions with proper citations of the previous methods in the field. The first known published method to perform separation of any signal in some latent continuous domain for neural networks was proposed in [D] and more similar to the contribution of this paper, the first method to formulate the separation problem to a classification-like problem where the targets are discrete latent vectors is in [C]. The authors use soundstream and it on their data which is simply a convolutional encoder that performs residual vector quantization identical to [C].
2. The authors should consider to compare against more modern state-of-the-art models in the source separation domain like Tiger [A] and TF-LocoFormer [B] and see if any of those models would perform better in terms of the separation of the signals.
3. Towards the point above an analysis that should be included a more holistic computational complexity analysis of the proposed model. Specifically, a potential issue of the proposed architecture is the missing evaluation in terms of the actual memory and computational requirements of the proposed architecture. I think the paper would be significantly benefited if the authors analyze the execution time of their model under some hardware, the actual memory footprint as well as the number of FLOPs. A complex architecture like the one proposed in here might severely enlarge the actual memory footprint of a forward pass of the network (or backward) mainly because of the intermediate activations tensors (for instance, this is generally true for recurrent and attention-based models that they seem to require less computation because of a smaller number of parameters but in real device deployment those models could be even 10 times slower to train than others). The authors should also report several other benchmark factors like actual memory requirements, time consuming on inference on GPU / CPU as the ones widely used in measuring the computational complexity of audio source separation [D] which is similar to the task solved in here.

I would gladly increase my score if the authors work properly to address the above issues to a proper degree.



[A] Xu, M., Li, K., Chen, G. and Hu, X., TIGER: Time-frequency Interleaved Gain Extraction and Reconstruction for Efficient Speech Separation. In The Thirteenth International Conference on Learning Representations.

[B] Saijo, K., Wichern, G., Germain, F.G., Pan, Z. and Le Roux, J., 2024, September. TF-Locoformer: Transformer with local modeling by convolution for speech separation and enhancement. In 2024 18th International Workshop on Acoustic Signal Enhancement (IWAENC) (pp. 205-209). IEEE.

[C] Jiang, Xilin. "Vector-quantized speech separation." (2021).

[D] Tzinis, E., Venkataramani, S., Wang, Z., Subakan, C. and Smaragdis, P., 2020, May. Two-step sound source separation: Training on learned latent targets. In ICASSP 2020-2020 IEEE International Conference on Acoustics, Speech and Signal Processing (ICASSP) (pp. 31-35). IEEE.

[E] Tzinis, E., Wang, Z., Jiang, X. and Smaragdis, P., 2022. Compute and memory efficient universal sound source separation. Journal of Signal Processing Systems, 94(2), pp.245-259.

**Questions:**

Is the second decimal point in the MSE metrics significant to report in these Tables? I doubt that any human being could tell the difference between 28.71 and 28.72 dB (actually I would argue that 27 and 28 would not even be differentiated properly). Are there any more useful statistics to report here?

---

> ### Author Response · Authors · 2025-11-21
>
> We thank the reviewer for taking the time to read our paper and for the provided feedback. In our response, we would like to address some of the issues raised in the review.
>
> > The authors should consider placing their contributions with proper citations of the previous methods in the field. The first known published method to perform separation of any signal in some latent continuous domain for neural networks was proposed in [D] and more similar to the contribution of this paper, the first method to formulate the separation problem to a classification-like problem where the targets are discrete latent vectors is in [C]. The authors use soundstream and it on their data which is simply a convolutional encoder that performs residual vector quantization identical to [C].
>
> We appreciate the reviewer pointing us to [C] and [D]. We were not aware of these references during our initial literature review and agree that they are relevant. We will reflect these works in the revised version of the paper. For now, we can indicate that our tokenizer is only inspired by SoundStream and uses Finite Scalar Quantization instead of Residual Vector Quantization. Furthermore, unlike [C], our tokenizer is completely pre-trained and serves only as a post-processing step to facilitate autoregressive prediction of our model. We do not exclusively use codebooks as part of our generation step.
>
> > The authors should consider to compare against more modern state-of-the-art models in the source separation domain like Tiger [A] and TF-LocoFormer [B] and see if any of those models would perform better in terms of the separation of the signals.
>
> We agree that comparing to recent audio source separation architectures is an interesting direction. In our work, we focused the setoup of more structured SOI, and we are interested in the exact match of the restored signal due to our BER metric. We are not sure how well the speech-focused models would perform in this domain.
>
> > Towards the point above an analysis that should be included a more holistic computational complexity analysis of the proposed model. Specifically, a potential issue of the proposed architecture is the missing evaluation in terms of the actual memory and computational requirements of the proposed architecture. I think the paper would be significantly benefited if the authors analyze the execution time of their model under some hardware, the actual memory footprint as well as the number of FLOPs. A complex architecture like the one proposed in here might severely enlarge the actual memory footprint of a forward pass of the network (or backward) mainly because of the intermediate activations tensors (for instance, this is generally true for recurrent and attention-based models that they seem to require less computation because of a smaller number of parameters but in real device deployment those models could be even 10 times slower to train than others). The authors should also report several other benchmark factors like actual memory requirements, time consuming on inference on GPU / CPU as the ones widely used in measuring the computational complexity of audio source separation [D] which is similar to the task solved in here.
>
> We agree that the current version underemphasizes computational complexity and memory usage, and we appreciate the concrete suggestions. One relevant point is that the tokenizer reduces the effective sequence length passed to the transformer (by a factor of 16 in our setup), which mitigates the quadratic cost of self-attention relative to operating directly at the waveform level.
>
> However, in this paper our primary focus was to show the proof of concept. While our models can be ran on a single GPU, we did not aggressively optimize for deployment efficiency. For reference, on an NVIDIA H100 96 GB GPU, running inference for the 5G model on 50 mixtures with batch size 10 takes about 25 seconds and uses approximately 22.5 GB of GPU memory. Other people in the field are focused on the practical application of the models we propose, and we did not work in that direction.
>
>
> > Is the second decimal point in the MSE metrics significant to report in these Tables? I doubt that any human being could tell the difference between 28.71 and 28.72 dB (actually I would argue that 27 and 28 would not even be differentiated properly). Are there any more useful statistics to report here?
>
> We agree that reporting MSE with two decimal places in dB can be too muсh. In our work, BER is the primary metric, because it directly reflects the performance of a digital communication system. MSE is used mostly in regimes where BER is already essentially zero, to distinguish how close the reconstruction is to an ideal reference. Moreover, while such a difference is indistinguishable by a human, a machine can detect such a difference, as it is the end consumer of the denoised signal.

---

### Official Review · Reviewer_1LTZ · 2025-11-01

**Soundness:** 2
**Presentation:** 3
**Contribution:** 2
**Rating:** 2
**Confidence:** 4

**Summary:**

This paper proposes a transformer-based model for radio-frequency (RF) signal separation, utilizing a SoundStream-style neural tokenizer to discretize input waveforms and training the transformer with cross-entropy loss over the quantized tokens. Evaluated on the MIT RF Challenge dataset, the model achieves improvements in bit error rate compared to prior architectures like WaveNet, but without introducing substantive architectural or theoretical novelty.

**Strengths:**

The paper tackles a practically relevant and challenging problem -- signal separation under RF interference. The use of cross-entropy loss on quantized token sequences, rather than waveform-level MSE, is technically appropriate for a discrete latent representation and improves compatibility with communication metrics such as BER. The writing and experimental presentation are clear and organized, helping reproducibility.

**Weaknesses:**

The model is a relatively straightforward adaptation of existing architectures (SoundStream tokenizer + transformer) to an RF dataset. Architectural, theoretical, or algorithmic innovations are limited.

Evaluation is restricted to the MIT RF Challenge dataset.

The paper provides no quantitative evidence on why each design choice (tokenization depth, transformer depth, etc.) matters, making the results difficult to interpret.

Assertions of robustness to unseen interference are based on synthetic, closely related test conditions, offering little evidence of real-world generalization.

Causality is discussed only in the appendix. Given the real-time nature of the application, it should instead be front and center.

The empirical results, though positive, are not compelling enough. The work appears better suited to an applied signal-processing or communications venue (e.g., ICASSP or SPAWC), where empirical improvements on established RF tasks are of primary interest.

**Questions:**

What happens when the signal experiences unknown channel fading? Can the model separate signals reliably under time-varying or unmodeled propagation effects?

Can the tokenizer and transformer be trained jointly, or are they frozen independently?

How sensitive is the system to quantization depth, token vocabulary size, or transformer depth?

Does the model generalize to modulation formats other than QPSK or to real-world RF mixtures beyond synthetic datasets?

---

> ### Author Response · Authors · 2025-11-21
>
> We thank the reviewer for taking time to read our paper and the provided feedback. In our response, we would like to address some of the issues raised in the review.
>
> > The model is a relatively straightforward adaptation of existing architectures (SoundStream tokenizer + transformer) to an RF dataset. Architectural, theoretical, or algorithmic innovations are limited.
>
> While the primary focus of our work is the source separation task, as well as the zero-shot generalization properties, we think our work has the following algorithmic contributions, motivated by the nature of source separation:
>
> - We choose the encoder-decoder transformer variant so that the decoder can both do autoregressive generation and use the processed mixture signal as a conditioning
> - Using causal cross-attention together with causal encoder makes the whole model causal
> - We propose a method for reconciliation of model outputs in case if the number of training samples in transformer’s window is less than the length needed on inference (see Appendix E)
>
> While the ideas of using tokenization and transformer backbone were existing in the past, our work utilizes those concepts for the task of source separation, where, to the best of our knowledge, conditional autoregressive generation of discrete tokens has not been used. In particular, SoundStream uses tokenizer for compression, while MusicLM performs autoregressive generation without a conditioning signal supported on similar time domain.
>
> > Evaluation is restricted to the MIT RF Challenge dataset.
>
> Due to limited computation resources, we focused on this dataset. In the future work, we would be interested in expanding our findings to other domains
>
> > The paper provides no quantitative evidence on why each design choice (tokenization depth, transformer depth, etc.) matters, making the results difficult to interpret.
>
> Before committing to training with tokenized representations, we have tried to use MSE objective and continuous tokenization. We have observed a strong improvement in 5G dataset from this design choice, which motivated us to explore discretized architectures more.
>
> | Method                         | MSE (dB) ↓ | log10(BER) ↓ |
> |--------------------------------|-----------:|-------------:|
> | RF transformer                 |     -46.32 |        -4.91 |
> | RF transformer (no tokenizer)  |     -10.23 |        -1.19 |
>
>
> We also do the ablation studies on the architecture of the tokenizer (see section 4.4).
>
> Tokenizer architecture ablation (final validation loss)
>
> | Quantizer | Transformer Blocks = 0 | Transformer Blocks = 4 |
> |----------|-------------------------|-------------------------|
> | FSQ      | 6.0e-05                | 2.0e-05                |
> | RVQ      | 2.0e-04                | 3.0e-04                |
>
> Signal length ablation
>
> | Input Signal Length N | Validation Loss |
> |------------------------|------------------|
> | 2,560                 | 2.5e-04          |
> | 40,960                | 1.0e-05          |
>
> In addition, we have performed ablation studies on the transformer depth on the CommSignal2:
>
> | SNR (dB) / MSE (db) |   2    |   4    |   8    |
> |---------:|:------:|:------:|:------:|
> |  -30.0   | -4.78  | -4.93  | -4.93  |
> |  -27.0   | -6.27  | -6.54  | -6.55  |
> |  -24.0   | -8.77  | -9.39  | -9.44  |
> |  -21.0   | -12.83 | -14.22 | -14.37 |
> |  -18.0   | -18.64 | -21.57 | -21.84 |
> |  -15.0   | -24.81 | -28.53 | -29.10 |
> |  -12.0   | -29.79 | -34.89 | -35.91 |
> |   -9.0   | -32.49 | -38.06 | -39.93 |
> |   -6.0   | -33.82 | -39.35 | -40.77 |
> |   -3.0   | -33.98 | -39.34 | -40.04 |
> |    0.0   | -33.64 | -39.34 | -40.70 |
>
> | SNR (dB) / log10(BER) |    2    |    4    |    8    |
> |---------:|:-------:|:-------:|:-------:|
> |  -30.0   | -0.62   | -0.63   | -0.63   |
> |  -27.0   | -0.78   | -0.81   | -0.81   |
> |  -24.0   | -1.05   | -1.11   | -1.11   |
> |  -21.0   | -1.49   | -1.61   | -1.62   |
> |  -18.0   | -2.17   | -2.37   | -2.38   |
> |  -15.0   | -2.94   | -3.13   | -3.18   |
> |  -12.0   | -3.82   | -4.01   | -4.07   |
> |   -9.0   | -4.63   | -5.41   | -5.41   |
> |   -6.0   | -5.41   |   -inf     | -5.41   |
> |   -3.0   | -4.93   | -5.11   | -4.93   |
> |    0.0   |   -inf    |   -inf    | -5.41   |
>
> > Assertions of robustness to unseen interference are based on synthetic, closely related test conditions, offering little evidence of real-world generalization.
>
> We have two main claims in robustness:
> - Our model, trained on mixtures of multiple structural interferences can match the performance of specialized models trained on individual interferences. This translates to the real world scenario, where the interference is unknown, but there are multiple possible options of what it could be
> - Gaussian noise is a common type of interference occurring in communication channel. Despite our model not being trained on Gaussian interference, the model is able to generalize to this setup, getting same performance as matched filter (which is designed specifically for Gaussian interference)

---

> > ### Author Response · Authors · 2025-11-21
> >
> > > Causality is discussed only in the appendix. Given the real-time nature of the application, it should instead be front and center.
> >
> > As the reviewer suggested, we will put more emphasis on causality in the final version of the paper, by adding the summary of this appendix section into the main part of the paper.
> >
> > > Can the tokenizer and transformer be trained jointly, or are they frozen independently?
> >
> > We train tokenizer, then freeze its weights, and train the transformer. The tokenizer’s weights affect the data on which the transformer is trained, so training tokenizer and transformer simultaneously would require more advanced techniques for gradient propagation. In other works, where generative models utilize tokenization of continuous data, such as MusicLM [3], MusicGen [4], and DALLE [5], the tokenizer is typically fixed beforehand, and then the transformer is trained. So, we used a similar two-stage pipeline in our case. Exploring the possibility of training the tokenizer and transformer jointly is a possible direction for future work.
> >
> > > Does the model generalize to modulation formats other than QPSK or to real-world RF mixtures beyond synthetic datasets?
> >
> > Our approach is generalizable to other modulation schemes. For each modulation scheme (which induces some structure on SOI) we need to train a separate tokenizer, after which we train the transformer. We do not use anything specific to QPSK to do the source separation.
> >
> >
> > [1] https://arxiv.org/abs/2107.03312
> >
> > [2] https://arxiv.org/abs/2301.11325
> >
> > [3] https://arxiv.org/pdf/2301.11325
> >
> > [4] https://arxiv.org/pdf/2306.05284
> >
> > [5] https://arxiv.org/pdf/2102.1209

---

### Meta-Review · Area_Chair_1Bu4 · 2026-01-03

**Summary:**

First, the reviewers found that there is limited novelty and architectural contribution to the proposed model. Reviewers suggested that the model is a straightforward adaptation of existing transformers to an RF dataset. These choices also lacked theoretical motivation and failed to cite a few prior works in discrete-domain source separation [1LTZ, FwN4, Cwjh, SA5U]. Next, another concern is the limited evaluation [1LTZ] and the need to compare against newer SOTA models [FwN4]. There should also be ablations to analyze the design choices [1LTZ] and the claims in training efficiency [SA5U]. Reviewers [FwN4] also ask for an analysis of computational complexity. Finally, reviewers [1LTZ, Cwjh] suggested that the paper is better suited for an applied engineering venue due to its focus on empirical improvements in the wireless communications domain.

**Reviewer Concerns:**

While some of the concerns regarding ablations and quantitative analysis may be addressed. The reviewers’ concerns regarding novelty and limited evaluation would largely be unaddressed in the AC opinion.

For example, Reviewer 1LTZ’s concerns on evaluating only on the MIT RF Challenge dataset. The authors responded:
> Due to limited computation resources, we focused on this dataset. In the future work, we would be interested in expanding our findings to other domains

This response is unlikely to have convinced the reviewer.

**Reviewer Scores:**

In the AC’s opinion, the reviewers are unlikely to change their score if they had been able to participate in the discussion. Some of the concerns are fundamental, and if not addressed in the first response, would be unlikely to be addressed in further discussion. Two of the reviewers have commented to keep their score, while the remaining don’t seem likely to raise it.

---

### Decision · Program_Chairs · 2026-01-26

Reject